# Revisiting Visual Understanding in Multimodal Reasoning through a Lens of Image Perturbation

## Abstract

Despite the rapid progress of multimodal large language models (MLLMs), the role of visual processing in multimodal reasoning remains underexplored. In a simple yet revealing experiment, we find that language-only models, when augmented with image captions, can sometimes outperform multimodal counterparts consuming raw visual inputs. This indicates that current MLLMs may perceive visual content but fail to effectively integrate it during reasoning. Moreover, even minimal visual perturbations such as small rotations lead to severe performance drops, exposing a fragility in their visual understanding. To address this overlooked bottleneck, we propose a lightweight visual perturbation (VP) framework that strengthens perceptual robustness without architectural changes or additional data. VP introduces three targeted dominance-preserving mixup, random rotation, and distractor concatenation which can be seamlessly integrated into post-training pipelines including SFT, DPO, and GRPO. Extensive experiments across four multimodal reasoning benchmarks show consistent absolute gains of 1–2 points, with improvements holding across datasets, training pipelines, and even advanced RL-tuned models. Ablation and task-level analyses further reveal how different perturbations uniquely benefit geometry, algebra, OCR, and chart reasoning. These findings underscore a central insight: better reasoning begins with better seeing.

## 1 Introduction

Recent advances in multimodal large language models (MLLMs) have led to impressive capabilities in vision-language understanding and reasoning (Liu et al., 2023; Zhu et al., 2023; Li et al., 2023; Wei et al., 2023; Wang et al., 2024b). Yet, their performance on math-centric reasoning tasks remains unsatisfactory, especially when visual information such as diagrams, charts, or spatial layouts is essential for problem-solving. Prior efforts have largely focused on two directions: (i) synthesizing large-scale multimodal datasets tailored for reasoning (Gao et al., 2023b; Zhang et al., 2024b; Dong et al., 2024), and (ii) advancing model architectures or training objectives (Meng et al., 2025; Peng et al., 2025; Deng et al., 2025; Wei et al., 2025b;a). However, a fundamental question has received little attention: how effectively do MLLMs process and integrate visual inputs during reasoning?

We begin with a simple yet striking observation. As illustrated in Figure 1, caption-augmented language models, in which an LLM is provided with captions generated by the same MLLM, can sometimes achieve comparable or even higher accuracy than the multimodal model itself. For example, on MathVision, Qwen2.5-7B with captions attains 28.8%, surpassing Qwen2.5-VL-7B at 25.6%. This suggests that while MLLMs can generate accurate visual descriptions, they often fail to leverage them for downstream reasoning. A second observation reinforces this diagnosis: applying benign perturbations such as random rotations, which preserve semantic content, causes large accuracy drops across multiple benchmarks; on MathVista, performance drops by 17.1 percentage points. Together, these results reveal a critical bottleneck. MLLMs are capable of perceiving images but do not robustly reason with visual information.

Motivated by this gap, we propose a lightweight visual perturbation (VP) framework that enhances perceptual robustness without introducing new data or modifying architectures. VP applies three targeted perturbations: dominance-preserving mixup, random rotation and distractor concatenation.

Figure 1: The left panel shows three settings: (i) Answer A (LLM-only), where a language model answers from text only; (ii) Answer B (MLLM), where a multimodal model jointly encodes both the question and the image; and (iii) Answer C (Caption-augmented LLM), where an image caption generated by the same MLLM is appended to the question for the LLM. The right panel presents quantitative results on MathVision (Wang et al., 2025b). We interestingly find that language-only models, when provided with image captions, can sometimes achieve even better performance than MLLMs that consume raw visual inputs. **This suggests that current MLLMs may generate accurate visual descriptions but fail to effectively integrate them during reasoning.**

These are designed to challenge models' ability to localize, filter, and reason over relevant visual features under structured variation. Crucially, VP is pipeline-agnostic and can be incorporated into existing alignment methods such as SFT (Tong et al., 2024), DPO (Rafailov et al., 2024), and GRPO (Guo et al., 2025b).

We conduct comprehensive experiments across four datasets and consistently observe absolute performance gains of 1–2 points on average across four benchmarks (MathVision, MathVista, MathVerse, and We-Math), demonstrating that our method yields robust improvements under diverse settings. It is worth noting that, unlike recent RL-tuned models that rely on collecting additional large-scale datasets to maximize performance, we deliberately restrict training to publicly available data to provide a more rigorous and controlled validation of our approach. Beyond dataset diversity, we also verify the effectiveness of our method across different alignment pipelines, including SFT, DPO, and GRPO, showing that the improvements hold regardless of the training pipeline. Importantly, our framework also yields improvements when built on current advanced models.

Through comprehensive ablation studies, we further highlight that not all visual perturbations are equally beneficial. Perturbations that preserve structural and semantic information while introducing meaningful difficulty lead to improvements, whereas perturbations that disrupt key visual cues consistently cause performance declines. To further understand why this occurred, we designed a follow-up analysis breaking down performance by problem type: geometry, algebra, table & chart, and OCR. A breakdown by problem type further shows that perturbations have task-specific effects. Rotation strongly benefits geometry reasoning (+6.8%) but reduces accuracy on algebra, table, and OCR tasks where text readability and spatial order are crucial. In contrast, dominance-preserving mixup and distractor concatenation provide more balanced gains, consistently improving algebra and OCR while maintaining competitive performance on geometry. These findings highlight that the effectiveness of perturbations depends on aligning them with the visual requirements of different reasoning tasks. These findings demonstrate that visual perturbation plays a critical role in multimodal mathematical reasoning—better reasoning fundamentally depends on better visual understanding.

The key contributions of this work are as follows:

- We identify a critical insight: caption-augmented LLMs can match or even surpass MLLMs, while minor image perturbations lead to significant accuracy declines, underscoring fundamental weaknesses in how current MLLMs process and integrate visual information.

- We propose a simple yet effective visual perturbation framework that consistently improves performance across datasets, pipelines, and model scales without requiring extra data;

- We provide detailed empirical and task-level analyses, showing how different perturbations complement each other and revealing the underexplored role of visual processing in multimodal reasoning.

## 2 RELATED WORK

**Multimodal Mathematical Reasoning.** The mathematical reasoning abilities of multimodal large language models (MLLMs) have become a central research focus (Zhuang et al., 2024; Gao et al., 2023a; Li et al., 2024; Dong et al., 2024; Hu et al., 2024; Yang et al., 2024b; Han et al., 2024; Guo et al., 2024). Compared to text-only reasoning (Luo et al., 2023; Yu et al., 2023), multimodal approaches must also process visual inputs, which makes tasks such as geometry and chart interpretation substantially more challenging (Chen et al., 2021). To address these challenges, prior work has mainly advanced along two directions. First, large-scale data synthesis and task-specific dataset construction have been widely explored, e.g., MAVIS for math-centric visual data generation (Zhang et al., 2024b), Math-LLaVA with MathV360K (Shi et al., 2024), Multimath with textbook data and GPT-4 validation (Peng et al., 2024), and reasoning-focused datasets such as LLaVA-CoT-100k (Xu et al., 2024) and Mulberry-260k (Yao et al., 2024). Second, architectural or algorithmic innovations have been proposed, including specialized encoders (Chen et al., 2024) and structured representations like R1-onevision (Yang et al., 2025). While these synthesis-driven and architecture-driven practices have led to significant progress, they rarely isolate and examine the role of visual processing itself. Notably, (Zhang et al., 2023) observe that the trustworthiness of results derived from the vision modality diminishes as tasks become more challenging, and they address this by introducing Vision Description Prompting—a technique that provides explicit image captions to improve performance on difficult vision-related tasks. This finding closely echoes our own observation in visual reasoning. And (Park et al., 2025) show that vision-language models exhibit a strong modality imbalance: models generalize from SIMPLE to HARD tasks well in the text modality but fail in the image modality. Their solution relies on image-via-text supervision and mixed-modality training to improve SIMPLE to HARD transfer on synthetic reasoning tasks. In contrast, our work focuses on strengthening the visual ability itself through simple visual perturbations, without requiring additional supervision or modality conversion.

**Multimodal Data Augmentation.** Data augmentation is a common strategy to improve multimodal models. MixGen (Hao et al., 2023) generates new image–text pairs by interpolating images and concatenating texts, while RobustMixGen (Kim et al., 2025) mitigates the spurious correlations in MixGen to enhance OOD robustness. Other approaches move beyond simple input mixing: XTRA (Gur et al., 2021) enriches training with retrieved image–caption pairs, and LEMDA (Liu et al., 2022) learns feature-level multimodal augmentations applicable across modalities. There is growing interest in using input perturbations to probe or improve VLM robustness. Verma et al. (Verma et al., 2024) systematically evaluate MLLMs under various distribution shifts and augmentations. Similarly, Wu et al. (Wu et al., 2024) propose NoiseBoost, which alleviates hallucinations in MLLMs through the integration of noise feature perturbations. While these methods demonstrate the value of visual perturbation, they are not specifically designed for visual reasoning tasks. In contrast, we design tailored visual perturbation strategies explicitly for such reasoning tasks and demonstrate their effectiveness when integrated with various MLLM training pipelines, including SFT (Tong et al., 2024), DPO (Rafailov et al., 2024), and GRPO (Guo et al., 2025a). Recently, concurrent work Noisyrollout (Liu et al., 2025a) propose a simple yet effective data augmentation method that mixes trajectories from both clean and moderately distorted images during RL training.

## 3 OBSERVATION

While multimodal reasoning has recently attracted significant research attention, the role of visual processing in MLLMs remains insufficiently explored. Our work begins with several simple yet revealing observations about how current MLLMs utilize visual information in reasoning tasks. First, we evaluate three settings as illustrated in Figure 1: (i) a pure language model evaluated directly on text-only questions, (ii) its multimodal counterpart that processes raw visual inputs, and (iii) the language model augmented with image captions generated by the same multimodal model (e.g., captions for Qwen2.5-7B are generated by Qwen2.5-VL-7B, and captions for QwQ-Preview are generated by Qwen2.5-VL-72B). We observe an interesting pattern in Table 1: pure language models, when provided with image captions, can sometimes achieve comparable or even better performance than multimodal models that process raw visual inputs.

Table 1: Performance of QwenLMs and Qwen-VLs on MathVision, MathVista, MathVerse, and We-Math benchmarks. Star symbol ($*$) denotes that LLMs are prompted with image captions generated by the same Qwen-VL for each question.

| Models | Size | MathVision | MathVista | MathVerse | We-Math |
|---|---|---|---|---|---|
| Qwen2.5-7B | 7B | 24.3 | 32.0 | 28.5 | 38.1 |
| Qwen2.5-VL-7B | 7B | 25.6 | 66.2 | 44.3 | 62.9 |
| Qwen2.5-7B$^*$ | 7B | **28.8** | 56.7 | 41.5 | 57.3 |
| QwQ-Preview | 32B | 37.3 | 34.5 | 34.1 | 41.8 |
| QvQ-Preview | 72B | 35.6 | 71.2 | 53.2 | 68.7 |
| QwQ-Preview$^*$ | 32B | **42.9** | 63.6 | **54.9** | 61.5 |
| *Benchmark w/ Random Rotation* | | | | | |
| Qwen2.5-VL-7B | 7B | 22.9 (**–2.7**) | 49.1 (**–17.1**) | 37.5 (**–6.8**) | 57.2 (**–5.7**) |

Specifically, on MathVision (Wang et al., 2024a), the 7B language model Qwen2.5-7B (Yang et al., 2024a) achieves a score of 24.3, nearly matching its multimodal counterpart Qwen2.5-VL-7B (Bai et al., 2025) at 25.6. Remarkably, when augmented with captions generated by Qwen2.5-VL-7B, Qwen2.5-7B improves to 28.8, surpassing Qwen2.5-VL-7B using raw visual input. This effect is not limited to small-scale models. QwQ-Preview (32B) achieves 37.3 on MathVision (Wang et al., 2025b), but rises to 42.9 with captions, exceeding the much larger 72B multimodal QvQ-Preview, which scores only 35.6. MathVerse (Zhang et al., 2024a) also demonstrates consistent results, reinforcing that this phenomenon is not confined to a single benchmark. Through this simple exploratory experiment, it suggests that current MLLMs might not effectively integrate their visual capabilities into reasoning tasks. We hypothesize that a caption-augmented language model establishes a natural lower bound for the performance of an ideal multimodal model on visual reasoning tasks, under the assumption that both models possess comparable language understanding capabilities. Since image captions are compressed representations of visual content, they inherently contain less information than the original images. Thus, a well-aligned and effective MLLM, which can directly access and process raw visual inputs, should in principle outperform or at least match a language model that only relies on generated captions. When this expectation is not met, it suggests that the MLLM may be underutilizing visual information or that its vision-language alignment is suboptimal.

To further probe the insufficient utilization of visual information in current MLLMs, we conduct robustness tests by applying controlled perturbations to the visual inputs. We deliberately begin with the simplest possible perturbation: random rotation. This transformation should not pose difficulty for a robust MLLM. As shown in Table 1, however, this simple change leads to severe degradation. For instance, Qwen2.5-VL-7B suffers a 17.1-point decline on MathVista, while similar drops are observed on MathVision, MathVerse and We-Math. Such consistent patterns across benchmarks indicate that the issue is systematic rather than benchmark-specific. These findings highlight that current MLLMs are fragile and sensitive to visual perturbations, reinforcing our earlier observation that they fail to effectively leverage raw visual inputs in reasoning tasks.

Based on these observations, it naturally reminds us to revisit the role of visual processing in multimodal reasoning. Rather than treating the fragility of current MLLMs under simple perturbations as a limitation to be avoided, we instead view it as an opportunity to better understand their reliance on visual inputs. By deliberately introducing structured variations, we can not only gain deeper insights into how MLLMs respond to different visual perturbations but also identify ways to strengthen the perceptual robustness.

## 4 VISUAL PERTURBATION STRATEGIES

Building upon our earlier observation (see Section 3) that MLLMs often underutilize visual information in multimodal reasoning, we propose a lightweight visual perturbation framework aimed at improving perceptual robustness. Our method involves applying controlled perturbations to input images that preserve core semantics while introducing visual variations. These perturbations are de-

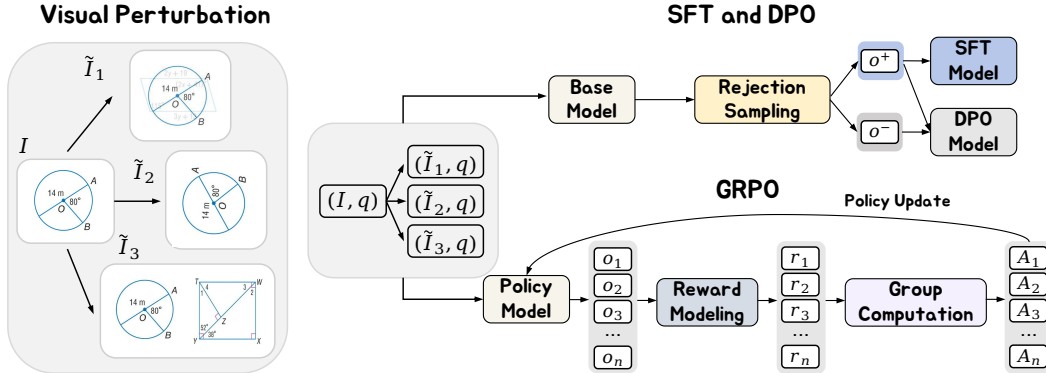

Figure 2: **Our visual perturbation (VP) framework consists of three strategies**: (1) dominance-preserving mixup that blends the input with a distractor using skewed alpha values, (2) random rotation that applies small angle rotations and (3) distractor concatenation that horizontally combines the input image with a random distractor. During training, these perturbations are applied across multiple alignment pipelines including SFT, DPO, and GRPO to enhance the model's perceptual robustness and reasoning consistency.

signed to challenge the model's ability to localize, extract, and reason over relevant visual information in the presence of noise, ambiguity, or structural shifts.

Importantly, we deliberately avoid complex perturbation designs, ensuring reproducibility while demonstrating that even simple perturbations can already yield significant gains.

In particular, we introduce three perturbation strategies at the image level, each targeting a different aspect of visual perception and reasoning. During training, one of the three perturbations is applied to each image uniformly at random unless otherwise specified.

**Dominance-Preserving Mixup.** Inspired by classic mixup (Zhang et al., 2017), we combine the original image $I$ and a distractor $I'$ using a skewed alpha-blending: $I_{\text{mix}} = \lambda I + (1 - \lambda)I'$, where $\lambda \in [0.75, 0.85]$ from an uniform distribution. Unlike standard mixup, our formulation preserves the dominant visual features of the original image while injecting low-level noise from an unrelated scene. This encourages the model to learn more invariant and robust visual features, focusing on the dominant structures relevant for reasoning rather than overfitting to low-level image textures or noise patterns.

**Random Rotation.** We randomly rotate the input image to simulate geometric transformations commonly encountered in real-world diagrams and figures. This perturbation is particularly valuable for geometry-centric problems, testing the model's spatial invariance and its ability to parse rotated structures or symbols.

**Distractor Concatenation.** Given an input image $I$, we horizontally concatenate a randomly sampled, distractor image $I'$, forming $[I; I']$. This strategy challenges whether the model can localize and attend to the relevant subregion of the visual input while ignoring irrelevant content. It mimics real-world settings where important information may appear alongside clutter, noise, or unrelated visual elements. Robust models should learn to suppress spurious visual signals and focus on the region aligned with the textual question.

## 5 EXPERIMENTS

### 5.1 EXPERIMENTAL SETUP

**Implementation Details.** We conduct experiments using Qwen2.5-VL-7B-Instruct (Bai et al., 2025) as our base model. For SFT and DPO training, we adopt the MS-Swift (Zhao et al., 2024b) framework, while for GRPO training we use the EasyR1 (Zheng et al., 2025) framework. For SFT and DPO, we first perform rejection sampling by generating 16 responses from Qwen2.5-VL-7B-Instruct for each instruction. The responses are evaluated for correctness by comparing the extracted answers with

Table 2: Performance comparison of GRPO training with and without visual perturbations (VP) across various training datasets. Results are evaluated on MathVision (Wang et al., 2025b), MathVista (Lu et al., 2023), MathVerse (Zhang et al., 2024a), and We-Math (Qiao et al., 2024) benchmarks. To make the performance gains clearer, we report the means and standard deviations over *three* runs. All values represent accuracy percentages (%). **VP consistently improves performance by 1–2 points, confirming its universality across datasets of different sizes and domains.**

| Model and Methods | Training Data | Benchmarks | | | | Average |
|---|---|---|---|---|---|---|
| | | **MathVision** | **MathVista** | **MathVerse** | **We-Math** | |
| Qwen2.5-VL-7B | – | 25.6 | 66.2 | 44.3 | 62.9 | 49.8 |
| GRPO | Geometry-3K | 27.07 ± 0.38 | 69.83 ± 0.49 | 46.70 ± 0.56 | 68.63 ± 0.85 | 53.10 ± 0.40 |
| **GRPO + VP** | Geometry-3K | 28.43 ± 0.75 | 72.63 ± 0.68 | 48.53 ± 0.45 | 70.17 ± 0.35 | **54.94 ± 0.35** |
| GRPO | MMR1-6K | 29.07 ± 0.21 | 70.00 ± 0.53 | 46.03 ± 0.60 | 68.97 ± 0.67 | 53.52 ± 0.23 |
| **GRPO + VP** | MMR1-6K | 31.20 ± 0.36 | 70.03 ± 0.61 | 47.63 ± 0.35 | 71.17 ± 0.42 | **55.01 ± 0.40** |
| GRPO | TQA-7K | 26.20 ± 0.66 | 69.17 ± 0.15 | 46.40 ± 0.26 | 66.43 ± 0.15 | 52.03 ± 0.15 |
| **GRPO + VP** | TQA-7K | 26.77 ± 0.21 | 71.60 ± 0.62 | 46.43 ± 0.15 | 67.77 ± 1.19 | **53.14 ± 0.23** |
| GRPO | GeoQA-8K | 27.30 ± 0.70 | 69.33 ± 0.32 | 46.93 ± 0.38 | 67.43 ± 1.27 | 52.77 ± 0.32 |
| **GRPO + VP** | GeoQA-8K | 27.77 ± 0.76 | 71.97 ± 0.55 | 48.80 ± 0.56 | 70.40 ± 0.53 | **54.73 ± 0.48** |

ground truth using Qwen2.5-32B-Instruct as the evaluator. For SFT, we select the longest correct response as the positive sample, training for 3 epochs with a learning rate of 1e-4 and weight decay of 0.1. For DPO, we choose both the longest correct response as the positive sample and the shortest incorrect response as the negative sample (Zhao et al., 2024a), training for 1 epoch with a learning rate of 5e-5, weight decay of 0.1, and warmup ratio of 0.05. For GRPO training, we follow the default hyperparameters in EasyR1, setting training episodes to 15, using AdamW optimizer with a learning rate of $1 \times 10^{-6}$, weight decay of $1 \times 10^{-2}$, and gradient clipping at a maximum norm of 1.0. We set the number of rollouts per episode to 5 for GRPO training. The vision tower of Qwen2.5-VL-7B is fine-tuned without freezing, and the GRPO objective incorporates a KL divergence penalty with a coefficient of 0.01 to stabilize training. During training, we adopt a simple accuracy-based reward function that assigns +1 for correct final answers and 0 for incorrect ones.

**Evaluation Benchmarks.** We evaluate the MLLMs on several multimodal mathematical reasoning benchmarks: MathVision (Wang et al., 2025b), MathVista (Lu et al., 2023), MathVerse (Zhang et al., 2024a), We-Math (Qiao et al., 2024). For more details about benchmarks, please see Appendix A.2. For all benchmarks, we prompt the models to place their final answers within a designated box format. We then employ Qwen2.5-32B-Instruct (Yang et al., 2024a) to evaluate answer correctness by comparing the extracted responses with ground truth answers, which often contain complex mathematical expressions. Note that our reported benchmark scores may differ from those in the original papers due to variations in evaluation protocols.

## 5.2 MAIN RESULTS

**Effectiveness Across Different Training Datasets.** Table 2 demonstrates the effectiveness of our visual perturbation (VP) framework under GRPO training across four diverse datasets (Geometry3K (Lu et al., 2021), MM-R1 (Leng, 2025), TQA (Kim et al., 2018), and GeoQA (Chen et al., 2021)). To more clearly capture the performance gains introduced by VP, we report the mean and standard deviation over three independent runs. Across all settings, incorporating VP leads to consistent accuracy gains of approximately 1–2 points over the vanilla GRPO baseline. Specifically, training on Geometry3K (3K samples) improves from 53.10% to 54.94%, on MM-R1 (6K samples) from 53.52% to 55.01%, on TQA (7K samples) from 52.03% to 53.14%, and on GeoQA (8K samples) from 52.77% to 54.73%. The improvements are consistent across datasets of varying sizes and domains, demonstrating the universality of VP as a training enhancement.

**Effectiveness Across Different Training Pipelines.** Table 3 evaluates the effectiveness of VP when applied to different alignment pipelines. We consider two commonly used training methods, supervised fine-tuning (SFT) and direct preference optimization (DPO), across four datasets. The

Table 3: Performance comparison of SFT and DPO training with and without visual perturbations (VP) across various training datasets. Results are evaluated on MathVision (Wang et al., 2025b), MathVista (Lu et al., 2023), MathVerse (Zhang et al., 2024a), and We-Math (Qiao et al., 2024) benchmarks.**VP consistently improves both SFT and DPO pipelines, demonstrating its pipeline-agnostic benefits.**

| Model and Methods | Training Data | MathVision | MathVista | MathVerse | We-Math | Average |
|---|---|---|---|---|---|---|
| Qwen2.5-VL-7B | – | 25.6 | 66.2 | 44.3 | 62.9 | 49.8 |
| SFT | Geometry-3K | 26.9 | 66.9 | 44.7 | 65.9 | 51.1 |
| SFT + VP | Geometry-3K | 27.5 | 68.6 | 45.9 | 67.5 | **52.4** |
| SFT | MMR1-6K | 27.9 | 67.3 | 46.3 | 64.9 | 51.6 |
| SFT + VP | MMR1-6K | 28.4 | 68.3 | 46.9 | 67.4 | **52.7** |
| SFT | TQA-7K | 25.2 | 68.6 | 46.7 | 63.3 | 51.0 |
| SFT + VP | TQA-7K | 26.5 | 71.5 | 47.1 | 65.1 | **52.6** |
| SFT | GeoQA-8K | 26.8 | 67.1 | 46.8 | 64.2 | 51.2 |
| SFT + VP | GeoQA-8K | 27.8 | 69.4 | 48.0 | 66.6 | **53.0** |
| DPO | Geometry-3K | 27.8 | 65.2 | 45.8 | 64.6 | 50.9 |
| DPO + VP | Geometry-3K | 28.9 | 70.7 | 47.5 | 67.5 | **52.7** |
| DPO | MMR1-6K | 28.3 | 67.7 | 46.2 | 65.8 | 52.5 |
| DPO + VP | MMR1-6K | 29.5 | 68.6 | 47.3 | 68.7 | **53.5** |
| DPO | TQA-7K | 26.2 | 70.7 | 47.3 | 65.5 | 52.4 |
| DPO + VP | TQA-7K | 27.2 | 72.8 | 49.0 | 66.4 | **53.9** |
| DPO | GeoQA-8K | 26.7 | 68.0 | 47.7 | 66.1 | 52.1 |
| DPO + VP | GeoQA-8K | 27.5 | 71.1 | 49.2 | 68.8 | **54.2** |

Table 4: Average accuracy (%) of different models trained with and without visual perturbations (VP). VP provides consistent gains across diverse architectures and datasets, confirming its role as a lightweight enhancement applicable to both baseline and advanced models. * indicates the results reported in their paper. We provide detailed results in Appendix A.3.

| Model | Training Dataset | Avg |
|---|---|---|
| MM-eureka-Qwen-7B | MMK12-16K | 52.5 54.1* |
| + VP | MMK12-16K | **54.3** |
| Qwen2.5-VL-7B | Geometry-3K | 53.1 |
| + VP | Geometry-3K | **54.7** |
| ThinkLite-VL-7B | ThinkLite-hard-11K | 54.2 57.1* |
| + VP | ThinkLite-hard-11K | **55.5** |
| VL-Rethinker-7B | ViRL-39K | 55.2 57.2* |
| + VP | ViRL-39K | **56.0** |

Table 5: Effect of different training data compositions with visual perturbations (VP). While moderate augmentation (e.g., Clean + 1× VP) yields the best improvements, excessive augmentation introduces redundancy and does not provide further gains.

| Training Mix | Size | Avg |
|---|---|---|
| All Clean | 2.1k | 53.1 |
| All VP | 2.1k | 54.3 |
| Half Clean + Half VP | 2.1k | 54.5 |
| Clean + 1x VP | 4.2k | **54.9** |
| Clean + 4x VP | 10.5k | 54.7 |

results show that VP consistently improves both pipelines. For SFT, the accuracy increases from 51.2% to 53.0% on average, while for DPO the performance increases from 52.1% to 54.2% on GeoQA-8K (Chen et al., 2021). Importantly, the gains appear across all four datasets rather than being confined to a specific training scenario, demonstrating that VP is a pipeline-agnostic enhancement. In summary, these findings confirm that VP is not tied to a particular optimization objective: whether models are trained via SFT or DPO, incorporating VP provides steady improvements and acts as a lightweight complement to existing training pipelines.

Table 6: The impact of data exposure. Training with 2 times Clean Data (simply repeating the clean data) does not bring any further performance improvement.

| Training Protocol | Training Epochs | Data Composition | MathVision | MathVista | MathVerse | We-Math | Avg |
|---|---|---|---|---|---|---|---|
| 1x Clean | 1x | 100% clean | 27.07 ± 0.38 | 69.83 ± 0.49 | 46.70 ± 0.56 | 68.63 ± 0.85 | 53.10 ± 0.40 |
| 2x Clean | 2x | 100% clean | 27.03 ± 0.65 | 68.90 ± 1.59 | 47.70 ± 0.25 | 68.80 ± 0.36 | 53.11 ± 0.45 |
| Clean + VP | 1x | 100% clean + 100% VP | 28.43 ± 0.75 | 72.63 ± 0.68 | 48.53 ± 0.45 | 70.17 ± 0.35 | 54.94 ± 0.35 |

**Complementary to Advanced Models.** We evaluate whether visual perturbations (VP) complement advanced models by continuing GRPO training from their released checkpoints on the same datasets, without introducing any new data. Concretely, MM-eureka-Qwen-7B (Meng et al., 2025), obtained by training Qwen2.5-VL-7B on the curated MMK12-16K (Meng et al., 2025) dataset, improves from 52.5% to 54.3% when further trained with VP. The same holds for ThinkLite-VL-7B (Wang et al., 2025c) (trained on ThinkLite-hard-11K (Wang et al., 2025c)), which rises from 54.2% to 55.5%, and for VL-Rethinker-7B (Wang et al., 2025a) (trained on ViRL-39K (Wang et al., 2025a)), which increases from 55.2% to 56.0%. Even the base Qwen2.5-VL-7B, trained only on Geometry-3K, benefits from VP, reaching 54.7%. These results confirm that VP acts as a lightweight and consistent enhancement, reliably adding 1–2 points on top of already strong models.

It is worth noting that the above advanced methods reach their high performance only large-scale data collection and sophisticated algorithmic designs. In sharp contrast, our approach uses only the publicly available Geometry-3K dataset (Lu et al., 2021)(2.1K samples): by simply adding VP, Qwen2.5-VL-7B attains 54.7% accuracy. This demonstrates that VP not only complements existing state-of-the-art pipelines but also serves as a universal and lightweight enhancer that narrows the gap between small-data baselines and advanced systems.

## 5.3 ABLATION STUDIES

**Different Data Compositions Using VP.** Table 5 analyzes the effect of varying the proportion of visual perturbation (VP) data during training. We find that simply replacing all clean data with VP yields moderate improvement (53.1% → 54.3%). A balanced mix of half clean and half VP achieves slightly higher gains (54.5%). The best result (54.9%) comes from combining the clean set with one additional VP-augmented version (i.e., doubling the data size). Interestingly, further increasing the number of VP variants (e.g., Clean + 4x VP, totaling 10.5k samples) does not lead to additional benefits and even shows marginal decline compared to the 1x setting. These results suggest that VP is most effective when used in moderation: a small amount of augmentation is sufficient to unlock its complementary benefits, while excessive perturbation introduces redundancy without yielding further gains.

**Impact of Data Exposure.** In this section, we want to emphasize that the performance gains of VP are not due to simply using more compute while it uses 2× the effective data exposure. That said, to ensure a fully fair comparison, we have conducted the controlled experiment: 2× Clean vs. Clean + VP. Specifically, we trained the model on the clean data for twice the number of epochs (i.e., 2 times training steps) to match the total effective data exposure in the Clean + VP setting. The experimental results in Table 6 demonstrate that: Training with 2 times Clean (simply repeating the clean data) does not bring any further performance improvement. This strongly validates that the performance gain originates from our carefully designed Visual Perturbation strategy (VP) and not from merely increasing the training steps or compute resources. We also want to highlight our motivation: When training data is limited, our goal is to unlock the full potential of existing data by exposing the model to diverse, semantics-preserving visual perturbations during training.

**Impact of Different Single Visual Perturbation.** We conduct a comprehensive ablation study on different perturbation strategies to evaluate their effect on multimodal reasoning performance with GRPO training on Geometry3K (Chen et al., 2021). Table 7 shows that different perturbations lead to markedly different outcomes. Dominance-Preserving Mixup increases it to 53.8 (+1.3%), Random Rotation lifts the average score from 53.1 to 54.1 (+1.9%) and Distractor Concatenation achieves the largest gain, reaching 54.3 (+2.3%). All three retain the complete original diagram while adding additional sources of interference, which encourages the model to improve spatial tolerance and its ability to ignore irrelevant content. Color Shift raises performance only slightly, from 53.1 to 53.2

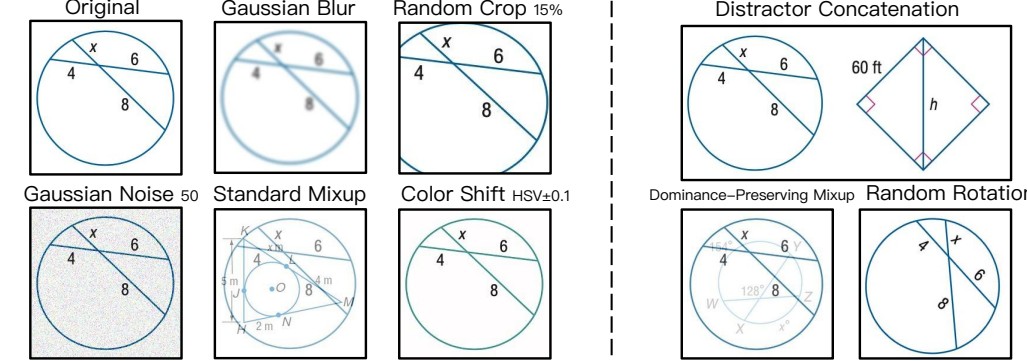

Figure 3: **Visualization of different perturbation strategies used in our ablation studies**. Specifically, Gaussian Blur, implemented with a medium kernel radius (2.5–7.5) that degrades fine details while maintaining visibility; Gaussian Noise, adding pixel-level noise with standard deviation 50, which disrupts low-level visual signals; Standard Mixup (0.45–0.55), blending two images with nearly equal weights; Color Shift, applying random offsets of ±0.1 to the hue and saturation channels in HSV space while preserving the original value (brightness) to maintain semantic content.

Table 7: Performance comparison of different perturbation strategies with GRPO training across mathematical benchmarks. All models are trained on the Geometry3K (Chen et al., 2021) dataset.

| Perturbation Type | Benchmarks | | | | Average |
|---|---|---|---|---|---|
| | MathVision | MathVista | MathVerse | We-Math | |
| None (Baseline) | 27.1 | 69.8 | 46.7 | 68.6 | 53.1 |
| Gaussian Blur | 24.9 | 69.5 | 40.7 | 60.8 | 49.0 (**–7.8%**) |
| Random Crop 15% | 25.4 | 70.6 | 39.8 | 65.8 | 50.4 (**–5.1%**) |
| Gaussian noise (std=50) | 27.1 | 69.3 | 44.1 | 67.2 | 51.9 (**–2.3%**) |
| Standard Mixup | 25.6 | 68.5 | 44.7 | 66.1 | 51.2 (**–3.6%**) |
| Color Shift (HSV ±0.1) | 27.7 | 69.6 | 47.6 | 67.9 | 53.2 (**+0.2%**) |
| Dominance-Preserving Mixup | 27.2 | 72.2 | 47.0 | 68.9 | 53.8 (**+1.3%**) |
| Random Rotation | 28.0 | 71.2 | 47.5 | 69.8 | 54.1 (**+1.9%**) |
| Distractor Concatenation | 28.4 | 70.6 | 47.7 | 70.3 | 54.3 (**+2.3%**) |

(+0.2%), because it modifies color appearance without introducing new distractor signals. In contrast, Gaussian Blur drops the average to 49.0 (–7.8%), Random Crop 15% to 50.4 (–5.1%), Standard Mixup to 51.2 (–3.6%), and Gaussian Noise (std=50) to 51.9 (–2.3%), as these perturbations damage or remove fine structural details such as edges, symbols, and layout boundaries that are essential for mathematical reasoning. Overall, perturbations that preserve structural and semantic information while introducing meaningful difficulty lead to improvements, whereas perturbations that disrupt key visual cues consistently cause performance declines.

## 5.4 QUALITATIVE ANALYSIS

To better understand how visual perturbations influence multimodal mathematical reasoning, we further analyze their effects across four representative problem categories—geometry, algebra, table & chart, and OCR-related tasks. While our analysis does not cover every problem type present in the benchmarks, these categories provide a diverse view of how perturbations interact with different reasoning demands.

**Geometry.** Rotation-based perturbations prove most effective here, improving accuracy from 146/381 to 172/381 (+6.8%). This suggests that forcing the model to reason about objects under varying orientations strengthens its spatial grounding. Dominance-preserving mixup and distractor concatenation also yield solid gains (+3.9% and +5.2%), showing that geometry tasks generally benefit from added visual variability.

Table 8: Impact of different perturbation strategies across problem types aggregated from all benchmarks. Each cell reports *correct / total* predictions and relative percentage change. Rotation benefits geometry reasoning but harms algebra, table, and OCR tasks, highlighting task-specific sensitivity to perturbations.

| Perturbation Type | Geometry | Algebra | Table & Chart | OCR |
|---|---|---|---|---|
| Baseline | 146/381 | 103/345 | 85/242 | 52/143 |
| Dominance-Preserving Mixup | 161/381 (**+3.9%**) | 117/345 (**+4.1%**) | 82/242 (**–1.2%**) | 55/143 (**+2.1%**) |
| Random Rotation | 172/381 (**+6.8%**) | 98/345 (**–1.4%**) | 80/242 (**–2.1%**) | 50/143 (**–1.4%**) |
| Distractor Concatenation | 166/381 (**+5.2%**) | 123/345 (**+5.8%**) | 96/242 (**+4.5%**) | 61/143 (**+6.3%**) |

**Algebra.** Unlike geometry, algebra tasks are harmed by random rotation (103/345 → 98/345, –1.4%), likely because rotations distort symbolic structures such as equations. In contrast, distractor concatenation (+5.8%) and mixup (+4.1%) both enhance performance, indicating that algebra problems benefit more from exposure to noisy but semantically consistent visual signals.

**Table & Chart.** Perturbations are more challenging in this category. Distractor concatenation improves accuracy from 85/242 to 96/242 (+4.5%), but mixup (82/242, –1.2%) and rotation (80/242, –2.1%) both degrade performance. This highlights that visual consistency and alignment are particularly important when models must parse structured tabular layouts.

**OCR.** For OCR-style tasks, distractor concatenation again provides the largest boost (52/143 → 61/143, +6.3%), while mixup yields a modest gain (+2.1%). Random rotation, however, slightly reduces accuracy (50/143, –1.4%), suggesting that text recognition remains highly sensitive to orientation changes.

## 6 DISCUSSION

In this work, our primary aim was to highlight the often-overlooked importance of visual processing in multimodal reasoning, and secondly, to demonstrate the surprising effectiveness of visual perturbation. We believe that, just as data augmentation has become a cornerstone in traditional vision tasks, the multimodal community should treat visual processing with the same level of rigor and attention. While we do not discount the remarkable progress driven by large-scale data collection and algorithm design; we argue that lightweight approaches such as visual perturbation, essentially a low-cost yet effective enhancement, deserve to be recognized as a community consensus and widely adopted.

Building on this work, we see substantial room for extension. The most immediate direction is to design more fine-grained perturbation strategies beyond the simple forms explored here, for example by adapting them to the characteristics of training images or dynamically aligning them with the training scheduler. In addition, visual perturbation should also be considered in conjunction with different algorithmic designs, where it may complement reinforcement learning, curriculum learning, or advanced alignment techniques.

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

# A  APPENDIX

## A.1  LLM USAGE

We used large language models (LLMs) as assistive tools in the preparation of this paper. Specifically, LLMs were employed for language editing and improving clarity. All research ideas, methodologies, theoretical results, and experiments were conceived and conducted by the authors. The authors take full responsibility for the content of this paper.

## A.2  EVALUATION BENCHMARKS

We evaluate the MLLMs on several multimodal mathematical reasoning benchmarks:

- **MathVision** (Wang et al., 2025b) is a challenging benchmark containing 3040 mathematical problems with visual contexts from real-world math competitions across 12 grades. It covers 16 subjects over 5 difficulty levels, including specialized topics like Analytic Geometry, Combinatorial Geometry, and Topology.
- **MathVista** (Lu et al., 2023) is a comprehensive benchmark for evaluating mathematical reasoning in visual contexts. It contains 1000 questions featuring diverse problem types including geometry, charts, and tables.
- **MathVerse** (Zhang et al., 2024a) is an all-around visual math benchmark designed for an equitable and in-depth evaluation of MLLMs. The test set contains 3940 multi-subject math problems with diagrams from publicly available sources, focusing on Plane Geometry and Solid Geometry.
- **We-Math** (Qiao et al., 2024) meticulously collect and categorize 1740 visual math problems in the test set, spanning 67 hierarchical knowledge concepts and 5 layers of knowledge granularity.

## A.3  DETAILED RESULTS IN TABLE 4

In this section, we provide the detailed per-benchmark results corresponding to Table 4 and offer some explanations.

First, the average accuracy of Qwen2.5-VL-7B+VP is 54.9 in our main experiment (Table 2), but decreases slightly to 54.7 when compared with advanced models. This difference arises because all results in Table 4 follow a two-stage training. For advanced models such as MM-eureka-7B (Meng et al., 2025), we apply VP by further training on their released checkpoints. To ensure a fair comparison, we apply the same two-stage setting to Qwen2.5-VL-7B, meaning that we continue training on perturbed data starting from the GRPO-trained checkpoint on Geometry-3K.

Second, the results of several advanced models differ from those originally reported in their papers. This discrepancy stems from using a different evaluation protocol. For all benchmarks, we prompt the models to place their final answers within a designated box format. We then employ Qwen2.5-32B-Instruct (Yang et al., 2024a) to efficiently evaluate answer correctness by comparing the extracted responses with ground truth answers, which often contain complex mathematical expressions. Variations in the evaluation model, system prompts, and other implementation details lead to differences from the reported numbers. Importantly, this paper does not aim to surpass any advanced models or claim state of the art performance; instead, our goal is to demonstrate the effectiveness of VP. As shown clearly in Table 9, under the same evaluation framework, VP consistently improves performance across all benchmarks.

Third, regarding the MathVista result of VL-Rethinker-7B, we were unable to obtain a reasonable score during reproducibility (its strong MathVision and MathVerse performance contrasts with an low MathVista accuracy). Since our focus is solely on examining the relative improvement brought by VP, we directly report the MathVista result from the original paper.

## A.4  ADDITIONAL RESULTS ON NATURAL IMAGES

In this section, we want to highlight that our focus is on improving the visual abilities of MLLMs, and we specifically choose visual reasoning because it is highly challenging. And we clarify that the mathematical reasoning task is only one task used to validate the effectiveness of Visual Perturbation

Table 9: Performance of different models with and without visual perturbations (VP). VP consistently improves accuracy across multiple benchmarks. * indicates results reported in the original papers.

| Model | Train Data | MathVision | MathVista | MathVerse | We-Math | Avg |
|---|---|---|---|---|---|---|
| MM-eureka-7B | MMK12-16K | 27.8 (26.9*) | 68.6 (73.0*) | 49.6 (50.3*) | 64.1 (66.1*) | 52.5 (54.1*) |
| + VP | MMK12-16K | 28.9 | 70.4 | 50.1 | 67.7 | **54.3** |
| Qwen2.5-VL-7B | Geometry-3K | 27.1 | 69.8 | 46.7 | 68.6 | 53.1 |
| + VP | Geometry-3K | 28.3 | 71.6 | 48.7 | 70.2 | **54.7** |
| ThinkLite-VL-7B | ThinkLite-11K | 28.3 (32.9*) | 71.9 (75.1*) | 48.4 (52.1*) | 68.1 | 54.2 (57.1*) |
| + VP | ThinkLite-11K | 30.0 | 73.3 | 49.0 | 69.8 | **55.5** |
| VL-Rethinker-7B | ViRL-39K | 29.3 (32.3*) | 74.9* | 49.5 (54.2*) | 67.2 | 55.2 (57.2*) |
| + VP | ViRL-39K | 30.2 | 74.9* | 50.4 | 68.6 | **56.0** |

(VP). To further demonstrate the effectiveness of VP, we provide follow-up experiments on non-math reasoning tasks here. We validate that our method is equally effective on natural images and conducted experiments using the SegZero (Liu et al., 2025b) training framework. We applied VP training to the VisionReasoner (Liu et al., 2025c) model on natural image segmentation tasks as shown in Table 10. It's worth noting that we do not fully optimize the performance on this specific task, yet the results still robustly demonstrate the effectiveness of VP. Our ultimate goal is for VP to become a consensus in the MLLM community, offering the possibility of further performance improvement without requiring additional data, or complex algorithmic and model structure design.

### A.5 REGARDING THE DIFFERENCES BETWEEN QWEN-LM AND QWEN-VL IN SECTION 3: OBSERVATION

In our observation, we hypothesize that a caption-augmented language model establishes a natural lower bound for the performance of an ideal multimodal model on visual reasoning tasks, under the assumption that both models possess comparable language understanding capabilities. Since image captions are compressed representations of visual content, they inherently contain less information than the original images. Thus, a well-aligned and effective MLLM, which can directly access and process raw visual inputs, should in principle outperform or at least match a language model that only relies on generated captions. When this expectation is not met, it suggests that the MLLM may be underutilizing visual information.

In the main content of our paper, we make a comparison between Qwen-VL (Answer B) and QwenLM (Answer C). As stated in Table 4 (Linguistic Performance) in the Qwen2.5-VL Technical Report (Bai et al., 2025), the LLM of Qwen-VL is initialized from the QwenLM backbone and its capabilities of the LLM model are maintained to be equivalent to the base QwenLM. For a fair comparison, we replace QwenLM with Qwen-VL here to get a new Answer C as shown in Table 11.

### A.6 SCALING BEHAVIOR OF VISUAL PERTURBATION

One may wonder whether the benefits of Visual Perturbation (VP) saturate quickly as the amount of augmented data increases. To address this concern, we conduct a systematic scaling study by training GRPO+VP models with increasing amounts of VP-augmented data sampled from GeoQA-8K. Specifically, we randomly select 2K, 4K, 6K, and 8K instances from GeoQA-8K, apply VP to generate corresponding perturbed image-question pairs, and fine-tune the model using the combined clean and perturbed data.

Table 10: Additional Results on Natural Images. Visual Perturbation shows consistent improvement build on VisionReasoner (Liu et al., 2025c).

| Method | ReasonSeg val | ReasonSeg test | RCO test | ARCO+ test | ARCOg test |
|---|---|---|---|---|---|
| VisionReasoner-7B (Liu et al., 2025c) | 66.3 | 63.6 | 78.9 | 74.9 | 71.3 |
| VisionReasoner-7B (Liu et al., 2025c) + VP | 67.1 | 65.2 | 80.4 | 74.1 | 72.9 |

Table 11: Comparison of Different Answer Variants.

| Method | Answer A | Answer B | Answer C | Answer C (New) |
|---|---|---|---|---|
| Proposed | 24.3 | 25.6 | 28.8 | 29.0 |

As shown in Table 12, the performance gain from VP consistently increases with more training data: With 2K instances, VP yields a +1.19% improvement (53.14 vs. 51.95). Scaling to 4K further improves the gain to +1.62% (54.30 vs. 52.68). At 6K, the gain slightly increases to +1.67% (54.47 vs. 52.80). Finally, with the full 8K set, VP achieves a +1.96% boost (54.73 vs. 52.77).

These results demonstrate that VP remains effective even at larger scales, with diminishing but positive returns. The consistent performance gap between VP and baseline models across all data sizes confirms that VP provides a stable and scalable robustness enhancement, rather than a marginal gain limited to small datasets.

Table 12: Scaling analysis of Visual Perturbation (VP) on GeoQA. All models are trained with GRPO. Reported scores are average accuracy (%).

| Training Set | Average Acc (%) | Gain from VP |
|---|---|---|
| GeoQA-2K | 51.95 | – |
| GeoQA-2K + VP | 53.14 | +1.19 |
| GeoQA-4K | 52.68 | – |
| GeoQA-4K + VP | 54.30 | +1.62 |
| GeoQA-6K | 52.80 | – |
| GeoQA-6K + VP | 54.47 | +1.67 |
| GeoQA-8K | 52.77 | – |
| GeoQA-8K + VP | 54.73 | +1.96 |

