# OpenReview forum: "Revisiting Visual Understanding in Multimodal Reasoning through a Lens of Image Perturbation"
_ICLR.cc/2026/Conference — Submitted to ICLR 2026_

### Official Review · Reviewer_EuC5 · 2025-10-24

**Soundness:** 2
**Presentation:** 3
**Contribution:** 2
**Rating:** 4
**Confidence:** 4

**Summary:**

This paper combines data augmentation techniques (rotation / mixup / concatenation) with VLM training (SFT/DPO/GRPO). The authors claim a uniform performance gain across training datasets / types on mathematical reasoning benchmarks. They also run ablations / qualitative analyses to analyze the effect of their perturbation method.

**Strengths:**

1. The motivation is clear and the idea is simple. It is easy to understand the position of the paper.
2. The experimental setup and methodology is very cleanly written. This enhances reproducibility of the results.
3. The paper reports mean, stddev across 3 runs in Table 2. This helps ensure statistical significance of the results.
4. Results are clearly annotated with the performance difference with the baseline in Tables 6, 7.

**Weaknesses:**

# Feedback on Experiment Setup / Results Analysis
- One of the biggest concerns I have about this paper is that Tables 2 and 3 do not seem to compare against a fair baseline. Based on the 54.9 number from Table 5, I can only guess that the numbers in Tables 2 and 3 are all from Clean + 1x VP. However, as the authors note, this is a number from when trained with a 2x compute. A fair comparison would be 1) 2x Clean vs. Clean + VP; or 2) Clean vs. half-Clean + half-VP. While I can believe that the proposed method will improve results on a fairly constructed experiment, I would suggest swapping out the main results with fair baselines, if the authors want to clearly argue the effectiveness of their method. At this point, it is unclear whether the numbers in Tables 2 and 3 are indeed from the proposed method (VP) or simply from more compute.
- I do not fully agree with the points in the paragraph named “Complementary to Advanced Models” (lines 350 - 366). First, I think the paragraph is confusing because it is simultaneously trying to convey two different messages: 1) training existing checkpoints with VP datasets further enhances performance; 2) Geometry3K+VP leads to “comparable” performance as SOTA models. For the point 1), it is unclear whether the performance boost comes from the proposed method VP or from more compute (or the quality of the base dataset). To correctly claim the performance gain as deriving from the proposed method, you should set the baseline to be existing checkpoints + GRPO on existing dataset (e.g., MM-eureka-Qwen-7B + GRPO on MMMK12-16K). For the point 2), first note that the Geometry-3K results are not from the same two-stage pipeline (GRPO on existing dataset -> GRPO on VP dataset), but from a one-stage pipeline (GRPO on Clean + VP). Therefore, the entry for Geometry-3K in Table 4 (lines 333-334) should be marked separately and the caption should also clarify that all other entries correspond to a two-stage pipeline of Clean vs. Clean->VP, whereas the Geometry-3K results correspond to Clean vs. Clean+VP. Also, it is unclear if you can claim Geometry3K+VP as being “comparable” to SOTA models. I believe you want to claim that Geometry3K+VP is comparable to ThinkLite-VL-7B or VL-Rethinker-7B from the numbers 54.2 < 54.9 <= 55.2. However, ThinkLite-VL-7B and VL-Rethinker-7B claim SOTA from a wider range of tasks (including non-math datasets like MMMU). If you train on a strictly math dataset, you will of course need a smaller dataset than training on a more comprehensive dataset to get the same performance gain on math benchmarks. Even then, I think you might be cherry-picking results to claim SOTA status. The results on MathVista, MathVerse, MathVision (removing We-Math which are not reported for ThinkLite-VL-7B and VL-Rethinker-7B) are 72.6, 48.5, 28.4 (avg 49.8). These numbers are all significantly lower than ThinkLite-VL-7B’s 75.1, 52.1, 32.9 (avg 53.4) or  VL-Rethinker-7B’s 74.9, 54.2, 32.2 (avg 53.8). The numbers are taken from the respective papers. So I can only guess that either the evaluation protocol was different from the papers or that either the Geometry3K dataset or the proposed VP method is particularly useful for We-Math. Unless you report on the detailed breakdown of the results (as you promise in line 344), I cannot make a judgment on this yet.
- I don’t necessarily agree with Line 177 (“assumption that both models possess comparable language understanding capabilities”). While the Qwen2.5-VL uses Qwen2.5 as the base LLM, the LLM weights have been further trained during the VLM integration, which may have changed certain benchmark performance (similar to Base vs. Instruct models on NLP / math benchmarks). It would be cleaner to compare against would be to use the exact same VLM (e.g., Qwen2.5-VL-7B) but ask the same question without the image


# Missing Citations to / Discussion on Previous Works
## Section 3 Observation 1 (LLM+caption > VLM)
The main argument here is that VLMs are able to extract enough information from an image to solve a relevant mathematical question even when it cannot directly solve it from the given image. A similar observation has already been made in previous literature. I recommend citing such results as relevant. Zhang et al. (2023) report that 1) VLMs can extract information from image even when it cannot solve the question in the image but can solve it in text format (Section 5.2); 2) prompting a VLM to extract relevant information from the image before solving a mathematical question boosts performance (Section 6). Park et al. (2025) also report that their finetuned models perform better when explicitly asked to convert an image to text first (Appendix I.6).

[1] Zhang et al., “Lost in Translation: When GPT-4V(ision) Can’t See Eye to Eye with Text
A Vision-Language-Consistency Analysis of VLLMs and Beyond,” arXiv preprint, 2023.
[2] Park et al., “Generalizing from SIMPLE to HARD Visual Reasoning: Can We Mitigate Modality Imbalance in VLMs?,” ICML 2025, 2025.

## Section 3 Observation 2 (benchmark performance drops when images are rotated)
VLMs underperforming when the input images are perturbed have also been explored by Verma et al. (2024). I would suggest citing the paper.

[3] Verma et al., “Evaluating Multimodal Large Language Models across Distribution Shifts and
Augmentations,” CVPR 2024, 2024.

## Section 5: Experimental Setup
Wu et al. (2024) also consider adding Gaussian noise to images when fine-tuning VLMs. While they only report evaluations for hallucinations, their result on resolving over-reliance on text tokens seems relevant to the work.

[4] Wu et al., “NoiseBoost: Alleviating Hallucination with Noise Perturbation for Multimodal Large Language Models,” arXiv preprint, 2024.


# Nit-picky details
1. There seems to be a typo for the MathVista numbers (either the original version or the perturbed version) for Qwen2.5-VL-7B in Table 1. The difference between 66.2 and 49.1 is 17.1, not 16.3.
2. On line 172, is there a reason you only point out MathVerse as the source of consistent results? Table 1 shows consistent results for MathVista and We-Math as well.
4. Line 212: just to clarify, it was chosen uniformly at random?
5. Line 240: how is the $lambda$ value chosen? What distribution is it drawn from?
6. There are multiple references where the first and last names are reversed (e.g., Zheng Yaowei in line 639)
7. Where is the table for detailed results that you promise in Line 344?

**Questions:**

- Can you elaborate on “higher-quality datasets such as Geometry3K and GeoQA-8K show more pronounced improvements” (line 311)? How do you define the quality here? Why do you consider Geometry3K and GeoQA-8K to be of higher quality than MMR1-6K or TQA-7K? Based on Tables 2 and 3, training on MMR1-6K leads to the best average accuracy across all training types (SFT/DPO/GRPO).
- I don’t know if I agree with line 185 (“This transformation [rotation] preserves all semantic
content”) Did you check if there are any explicit references to spatial relations between objects in the image (e.g., “left/right/up/right”) in the questions in the existing benchmarks? If so, the question might no longer be correct after the images are rotated.
- Line 214: How did you determine the distractor is semantically irrelevant? Did you use an external VLM? Did you hand-curate such data points? Is there any reference (e.g., “point A”) that appears in both images?
- What is the rationale behind choosing the longest/shortest response for SFT/DPO?

---

> ### Author Response · Authors · 2025-11-23
> **Response to Reviewer EuC5 (1/5)**
>
> Thank you very much for the time spent, the detailed  review and feedback, particularly regarding some of the less rigorous claims and minor errors in our article. This has been extremely helpful to us, and **we sincerely thank the reviewer once again**. We would like to address your concerns below.
>
> ---
>
> **Q1**: “One of the biggest concerns I have about this paper is that Tables 2 and 3 do not seem to compare against a fair baseline. Based on the 54.9 number from Table 5, I can only guess that the numbers in Tables 2 and 3 are all from Clean + 1x VP. However, as the authors note, this is a number from when trained with a 2x compute. A fair comparison would be 1) 2x Clean vs. Clean + VP; or 2) Clean vs. half-Clean + half-VP. While I can believe that the proposed method will improve results on a fairly constructed experiment, I would suggest swapping out the main results with fair baselines, if the authors want to clearly argue the effectiveness of their method. At this point, it is unclear whether the numbers in Tables 2 and 3 are indeed from the proposed method (VP) or simply from more compute.”
>
> **A1**: Thank you for raising this important point. **We clarify that the numbers in Tables 2 and 3 are indeed trained under the condition of Clean + 1x VP. While this results in 2× the effective data exposure, we want to emphasize that the performance gains reported are not due to simply using more compute.**
>
> That said, to ensure a fully fair comparison, **we have now conducted the suggested controlled experiment**: 2× Clean vs. Clean + VP. Specifically, we trained the model on the clean data for twice the number of epochs (i.e., 2 times training steps) to match the total effective data exposure in the Clean + VP setting.
>
> The experimental results demonstrate that: **Training with 2 times Clean (simply repeating the clean data) does not bring any further performance improvement.** This strongly validates that the performance gain originates from our carefully designed Visual Perturbation strategy (VP) and not from merely increasing the training steps or compute resources.
> **We will integrate this crucial control experiment and its conclusion into our revised manuscript to ensure maximum rigor and clarify our claim.**
> We also want to highlight our motivation: When training data is limited, our goal is to unlock the full potential of existing data by exposing the model to diverse, semantics-preserving visual perturbations during training.
>
> |Training Protocol |Training Epochs |Data Composition |MathVision|MathVista|MathVerse|WeMath| Average Acc (%)|
> |---|---|---|---|---|---|---|---|
> |1x Clean |1x |100% clean |27.07 ± 0.38 | 69.83 ± 0.49 | 46.70 ± 0.56 | 68.63 ± 0.85 | 53.10 ± 0.40|
> |2× Clean |2× |100% clean |27.03 ± 0.65 | 68.90 ± 1.59 | 47.70 ± 0.25 | 68.80 ± 0.36 | 53.11 ± 0.45 |
> |Clean + VP|1× |100% clean + 100% VP |28.43 ± 0.75 |72.63 ± 0.68|48.53 ± 0.45|70.17 ± 0.35|54.94 ± 0.35 |

---

> > ### Author Response · Authors · 2025-11-23
> > **Response to Reviewer EuC5 (2/5)**
> >
> > **Q2**: “I do not fully agree with the points in the paragraph named “Complementary to Advanced Models” (lines 350 - 366). First, I think the paragraph is confusing because it is simultaneously trying to convey two different messages: 1) training existing checkpoints with VP datasets further enhances performance; 2) Geometry3K+VP leads to “comparable” performance as SOTA models. For the point 1), it is unclear whether the performance boost comes from the proposed method VP or from more compute (or the quality of the base dataset). To correctly claim the performance gain as deriving from the proposed method, you should set the baseline to be existing checkpoints + GRPO on existing dataset (e.g., MM-eureka-Qwen-7B + GRPO on MMMK12-16K). For the point 2), first note that the Geometry-3K results are not from the same two-stage pipeline (GRPO on existing dataset -> GRPO on VP dataset), but from a one-stage pipeline (GRPO on Clean + VP). Therefore, the entry for Geometry-3K in Table 4 (lines 333-334) should be marked separately and the caption should also clarify that all other entries correspond to a two-stage pipeline of Clean vs. Clean->VP, whereas the Geometry-3K results correspond to Clean vs. Clean+VP.”
> >
> > **A2**:  Thank you for your detailed feedback, which ensures the rigor of our experiments regarding baselines and the consistency of comparison.
> >
> > First, as discussed in A1, we have demonstrated that the performance boost is not due to more compute.
> >
> > Second, regarding Table 4, MM-eureka-Qwen-7B is a model already trained using GRPO on MMK12-16K. MMK12-16K is a dataset proposed and collected in their paper, and this dataset was used to obtain MM-eureka-Qwen-7B via GRPO training. Similarly, ThinkLite-hard-11K and ViRL-39K are also datasets proposed by the corresponding models, which were used to obtain ThinkLite-VL-7B and VL-Rethinker-7B via training. In Table 4, we intended to show that applying VP using their respective datasets still yields further performance improvement on these already advanced models. **Thus, the baseline MM-eureka-Qwen-7B Acc 52.5 is the result obtained by applying GRPO Training on MMK12-16K to the existing checkpoint (Qwen2.5-VL-7B-Instruct)**. Likewise, the Qwen2.5-VL-7B Acc 53.1 is obtained by applying GRPO Training on Geometry3K to the existing checkpoint (Qwen2.5-VL-7B-Instruct). The results for ThinkLite-VL-7B and VL-Rethinker-7B follow the same logic. Therefore, the comparison presented here is fair.
> > Regarding the results after adding VP, we acknowledge that Geometry3K + VP was a one-stage result, while ThinkLite-VL-7B + VP was a two-stage result. The comparison here was indeed unfair. **Therefore, following your suggestion, we conducted a two-stage experiment to ensure a fair comparison with all other existing checkpoint + VP results**. We have already replaced the Geometry3K + VP result in the paper with the new two-stage result in the revised manuscript.
> > This clarifies the two practical use cases for VP: 1. From-scratch training on a new dataset (Clean + VP together), and 2. Continual improvement of an existing checkpoint using perturbed data.
> >
> > |Model| Training Dataset| Average Acc|
> > |---|---|---|
> > | MM-eureka-Qwen-7B | MMK12-16K | 52.5 |
> > | + VP | MMK12-16K | 54.3 |
> > | Qwen2.5-VL-7B on Geo3k | Geometry-3K | 53.1 |
> > | + VP |  Geometry-3K |  54.7 |
> > |ThinkLite-VL-7B | ThinkLite-hard-11K | 54.2 |
> > |+ VP | ThinkLite-hard-11K | 55.5 |
> > |VL-Rethinker-7B |  ViRL-39K | 55.2 |
> > | + VP | ViRL-39K | 56.0 |
> >
> > [1] Meng, Fanqing, et al. "Mm-eureka: Exploring the frontiers of multimodal reasoning with rule-based reinforcement learning." arXiv preprint arXiv:2503.07365 (2025).
> >
> > [2] Wang, Xiyao, et al. "Sota with less: Mcts-guided sample selection for data-efficient visual reasoning self-improvement." NeurIPS 2025
> >
> > [3] Wang, Haozhe, et al. "Vl-rethinker: Incentivizing self-reflection of vision-language models with reinforcement learning." NeurIPS 2025

---

> > > ### Author Response · Authors · 2025-11-23
> > > **Response to Reviewer EuC5 (3/5)**
> > >
> > > **Q3**: “Also, it is unclear if you can claim Geometry3K+VP as being “comparable” to SOTA models. ... ThinkLite-VL-7B and VL-Rethinker-7B claim SOTA from a wider range of tasks (including non-math datasets like MMMU). ... Even then, I think you might be cherry-picking results to claim SOTA status. ... ” (character limited)
> > >
> > > **A3**: Thank you for your comments. I would like to clarify these points blow.
> > >
> > > First, **we acknowledge that the claim "Comparable to SOTA models" was not a rigor claim, and we have immediately revised it in the updated manuscript**. **We want to clarify that our intention was never to perform a definitive comparison with the final performance of SOTA methods; we only wished to demonstrate the effectiveness of VP and conduct a relative comparison of how effective VP is**. Our goal was not to surpass SOTA or claim final SOTA comparability. This claim clearly caused misinterpretation, and we have therefore removed it.
> > >
> > > Second, models like ThinkLite-VL-7B and VL-Rethinker-7B achieve optimal performance through meticulously collected specialized datasets. The VL-Rethinker paper states that they used "novel data collected from the web." **ThinkLite-VL publicly disclosed that their collected ThinkLite-70K dataset includes Geometry3K (as shown in Figure 3 of their paper)**, while VL-Rethinker did not publicly disclose theirs. Therefore, even though we only train on mathematical datasets, these SOTA models have a distinct advantage in mathematical benchmarks due to their access to broader, specialized datasets. We conducted an additional experiment to prove this point, comparing the training results on the more diverse ThinkLite-70K dataset versus the math-only Geo3K dataset:
> > >
> > > |Datasets | MathVision| MathVista | MathVerse| WeMath | Avg |
> > > |---|---|---|---|---|---|
> > > | Geometry3k | 27.1 | 69.8 | 46.7 | 68.6| 53.1 |
> > > | ThinkLite70k | 26.9 | 71.2 | 50.6 | 68.7 | 54.4 |
> > >
> > > The results clearly show that training on the more diverse ThinkLite-70K yields superior results compared to training solely on Geometry3k.
> > > Finally, **we assert that we have not engaged in cherry-picking.** The entire premise of our paper is centered on making a fair and clean comparison, including conducting standard deviation experiments and strictly using public datasets rather than collecting our own. Given that our starting point was not to claim SOTA, cherry-picking results would be meaningless to our objective. To ensure full transparency, we disclose the complete test results of advanced models tested on our evaluation code:
> > >
> > > |model | MathVision| MathVista | MathVerse| WeMath | Avg |
> > > |---|---|---|---|---|---|
> > > |MM-eureka-7B| 27.8 | 68.6 | 49.6 | 64.1 | 52.5 |
> > > |ThinkLite-7B | 28.3 | 71.9 | 48.4 | 68.1 | 54.2 |
> > > |VL-Rethinker-7B | 29.3 | 74.9*(reported in VL-Rethinker) | 49.5 | 67.2 | 55.2 |
> > >
> > > **Admittedly, there are differences between the results we tested and the results reported in their original papers. However, we want to state that this phenomenon is very common[1,2,3,4,5]. For example, in Table 2 of VL-Rethinker [1], the accuracy for OpenVL-Thinker-7B[3] on MathVista and MathVerse is 70.2 and 47.9, respectively, whereas the original OpenVL-Thinker paper reports 72.3 and 50.3. Similarly, 16 results in Table 2 of the ThinkLite[2] paper do not align with their original papers. Specifically, in Table 3 of Shuffle-r1 [5], they report the accuracy of ThinkLite-VL-7B **is** 28.0, 72.4, 45.2, and 69.3 for MathVision, MathVista, MathVerse, and WeMath, respectively. And the accuracy of VL-Rethinker-7B is 29.7, 72.0, 51.7, 70.1.**
> > >
> > > In fact, **we attempted to align the results**, but we found that even after aligning the system prompt and other evaluation settings, we could only reproduce their results using their own evaluation code. However, that same code framework often shows discrepancies(sometimes large) when testing other models compared to their reported results. Therefore, we chose to use our own lightweight, evaluation code(already public) framework (using Qwen2.5-32B) to unify and ensure a fair relative comparison (with VP vs. without VP) across all models.
> > > **Since we have no intention of claiming any SOTA performance, we nerver mind simply reporting their original paper's results in our experiments.**
> > >
> > > [1] Wang, Haozhe, et al. "Vl-rethinker: Incentivizing self-reflection of vision-language models with reinforcement learning." NeurIPS 2025
> > >
> > > [2] Wang, Xiyao, et al. "Sota with less: Mcts-guided sample selection for data-efficient visual reasoning self-improvement." NeurIPS 2025
> > >
> > > [3] Deng, Yihe, et al. "Openvlthinker: An early exploration to complex vision-language reasoning via iterative self-improvement." NeurIPS 2025
> > >
> > > [4] Wan, Zhongwei, et al. "Srpo: Enhancing multimodal llm reasoning via reflection-aware reinforcement learning." NeurIPS 2025
> > >
> > > [5] Zhu, Linghao, et al. "Shuffle-r1: Efficient rl framework for multimodal large language models via data-centric dynamic shuffle." arXiv preprint arXiv:2508.05612 (2025).

---

> > > > ### Author Response · Authors · 2025-11-23
> > > > **Response to Reviewer EuC5 (4/5)**
> > > >
> > > > **Q4**: "I don’t necessarily agree with Line 177 (“assumption that both models possess comparable language understanding capabilities”). While the Qwen2.5-VL uses Qwen2.5 as the base LLM, the LLM weights have been further trained during the VLM integration, which may have changed certain benchmark performance (similar to Base vs. Instruct models on NLP / math benchmarks). It would be cleaner to compare against would be to use the exact same VLM (e.g., Qwen2.5-VL-7B) but ask the same question without the image"
> > > >
> > > > **A4**: Thank you for this insightful suggestion. We would like to address this by first explaining the rationale behind our original experimental design and then presenting the new results following your advice.
> > > >
> > > > First, regarding our motivation, we hypothesize that a caption-augmented language model establishes a natural lower bound for the performance of an ideal multimodal model on visual reasoning tasks, under the assumption that both models possess comparable language understanding capabilities. Since image captions are compressed representations of visual content, they inherently contain less information than the original images. **Thus, a well-aligned and effective MLLM, which can directly access and process raw visual inputs, should in principle outperform or at least match a language model that only relies on generated captions.** When this expectation is not met, it suggests that the MLLM may be underutilizing visual information.
> > > >
> > > > Second, we believe our original comparison between Qwen-VL (Answer B) and QwenLM (Answer C) was reasonable under this hypothesis. **As stated in Table 4 (Linguistic Performance) in the Qwen2.5-VL Technical Report [1], the LLM of Qwen-VL is initialized from the QwenLM backbone and its capabilities of the LLM model are maintained to be equivalent to the base QwenLM.**
> > > >
> > > > Finally, we fully agree with you that replacing QwenLM with Qwen-VL in the Answer C setup provides a more rigorous control to strictly isolate the "integration" factor. As suggested, **we conducted the additional experiment where we feed the exact same "Question + Caption" input to Qwen-VL to get a new Answer C**. As shown in the table below, even when using the exact same model, the setup with generated captions (Answer C - New) still outperforms the image-only setup (Answer B).
> > > >
> > > > |Answer A|Answer B|Answer C|Answer C (New)|
> > > > | --- | --- | --- | --- |
> > > > |24.3|25.6|28.8| 29.0 |
> > > >
> > > > [1] Bai, Shuai, et al. "Qwen2. 5-vl technical report." arXiv preprint arXiv:2502.13923 (2025).
> > > >
> > > > ---
> > > > **Q5**: Discussion on Previous Works
> > > >
> > > > **A5**: Thank you for the suggestion. We have incorporated the discussion of these works into our Related Work section.
> > > >
> > > > ---
> > > >
> > > > **Q6**: There seems to be a typo for the MathVista numbers (either the original version or the perturbed version) for Qwen2.5-VL-7B in Table 1. The difference between 66.2 and 49.1 is 17.1, not 16.3.
> > > >
> > > > **A6**: Thank you for the reminder. We have already corrected this point in the revised manuscript.
> > > >
> > > > ---
> > > > **Q7**: On line 172, is there a reason you only point out MathVerse as the source of consistent results? Table 1 shows consistent results for MathVista and We-Math as well.
> > > >
> > > > **A7**: Thank you for the reminder. We intended to express that MathVerse also showed consistent results and mistakenly left out "also." We have corrected this point in the revised manuscript.
> > > >
> > > > ---
> > > >
> > > > **Q8**: Line 212: just to clarify, it was chosen uniformly at random?
> > > >
> > > > **A8**: Yes, it was chosen uniformly at random. The three perturbations are applied randomly to the image with equal probability. We have added this detail to the revised manuscript.
> > > >
> > > > ---
> > > >
> > > > **Q9**: Line 240: how is the lambda value chosen? What distribution is it drawn from?
> > > >
> > > > **A9**: The lambda ($\lambda$) value, which serves as the mixing coefficient for the Dominance-Preserving Mixup perturbation, is chosen by drawing from a **Uniform distribution**. In our implementation, $\lambda$ is sampled uniformly at random within the specified minimum and maximum bounds: $\lambda \sim \text{Uniform}(\lambda_{\text{min}}, \lambda_{\text{max}})$, implemented via lam = np.random.uniform(lambda_min, lambda_max). In this work, $\lambda_{\text{min}}$ is set to 0.75 and $\lambda_{\text{max}}$ is set to 0.85. We note that, for the sake of simplicity, **we did not perform further detailed hyperparameter tuning on this range**. We have added this detail to the revised manuscript.
> > > >
> > > > ---
> > > >
> > > > **Q10**: There are multiple references where the first and last names are reversed (e.g., Zheng Yaowei in line 639)
> > > >
> > > > **A10**: Thank you for the reminder. We have already corrected this in the revised manuscript.

---

> > > > > ### Author Response · Authors · 2025-11-23
> > > > > **Response to Reviewer EuC5 (5/5)**
> > > > >
> > > > > **Q11**: Where is the table for detailed results that you promise in Line 344?
> > > > >
> > > > > **A11**:
> > > > > Thanks for point this out. We forgot to include this detailed table. Since the full table occupies too much space in the main text, we should have placed it in the Appendix but unfortunately, we overlooked this necessity. The complete table is provided below, and we have now added it to the Appendix in the revised manuscript.
> > > > > |Model and Methods |Training Data |MathVision |MathVista |MathVerse |We-Math |Average |
> > > > > |---|---|---|---|---|---|---|
> > > > > |Qwen2.5-VL-7B|– |25.6|66.2|44.3|62.9|49.8|
> > > > > |SFT | Geometry-3K | 26.9 | 66.9 | 44.7 | 65.9 |51.1|
> > > > > |SFT + VP | Geometry-3K|27.5| 68.6 | 45.9 | 67.5 |52.4|
> > > > > |SFT | MMR1-6K |27.9 | 67.3 | 46.3 | 64.9 |51.6|
> > > > > |SFT + VP | MMR1-6K| 28.4 | 68.3 | 46.9 | 67.4|52.7|
> > > > > |SFT | TQA-7K |25.2 | 68.6 | 46.7 | 63.3 |51.0|
> > > > > |SFT + VP | TQA-7K |26.5 | 71.5 |47.1 |65.1 |52.6|
> > > > > |SFT | GeoQA-8K | 26.8 | 67.1|  46.8 | 64.2  |51.2|
> > > > > |SFT + VP | GeoQA-8K |27.8 | 69.4 | 48.0 |66.6|53.0|
> > > > > |DPO | Geometry-3K |27.8 |65.2 |45.8 |64.6 |50.9|
> > > > > |DPO + VP | Geometry-3K|28.9 | 70.7 |47.5| 67.5 |52.7|
> > > > > |DPO | MMR1-6K |28.3 |67.7 | 46.2 | 65.8|52.5|
> > > > > |DPO + VP | MMR1-6K| 29.5 | 68.6 | 47.3 | 68.7|53.5|
> > > > > |DPO | TQA-7K |26.2 |70.7 |47.3 |65.5 |52.4|
> > > > > |DPO + VP | TQA-7K |27.2 |72.8 |49.0| 66.4 |53.9|
> > > > > |DPO | GeoQA-8K |26.7 | 68.0 | 47.7 | 66.1 |52.1|
> > > > > |DPO + VP | GeoQA-8K |27.5 | 71.1 |49.2 |68.8|54.2|
> > > > >
> > > > >
> > > > > ---
> > > > >
> > > > > **Q12**: Can you elaborate on “higher-quality datasets such as Geometry3K and GeoQA-8K show more pronounced improvements” (line 311)? How do you define the quality here? Why do you consider Geometry3K and GeoQA-8K to be of higher quality than MMR1-6K or TQA-7K? Based on Tables 2 and 3, training on MMR1-6K leads to the best average accuracy across all training types (SFT/DPO/GRPO).
> > > > >
> > > > > **A12**:Thank you for pointing out this loosely speaking. The term "high quality" here was indeed very imprecise, and we have removed this phrasing and clarified the underlying mechanism in the revised manuscript.
> > > > >
> > > > > ---
> > > > >
> > > > > **Q13**: I don’t know if I agree with line 185 (“This transformation [rotation] preserves all semantic content”) Did you check if there are any explicit references to spatial relations between objects in the image (e.g., “left/right/up/right”) in the questions in the existing benchmarks? If so, the question might no longer be correct after the images are rotated.
> > > > >
> > > > > **A13**: Thank you for pointing out this lack of rigor in our paper. Your concern regarding absolute spatial references (e.g., "left/right") is entirely valid, and we agree that the claim "This transformation preserves all semantic content" (Line 185) is an overstatement; we will immediately revise the phrasing in the updated manuscript. We would like to clarify, however, that in the context of our mathematical benchmarks, the core semantic content required by the questions is generally relational and mathematically invariant, meaning values and relationships remain unchanged after rotation. Furthermore, we have verified that the vast majority of questions in these benchmarks do not rely on absolute spatial positioning (e.g., "What is the number in the top-left corner?"). Therefore, while the rotation operation may technically violate absolute spatial semantics, it successfully preserves the core relational semantics essential for the mathematical reasoning task.
> > > > >
> > > > > ---
> > > > >
> > > > > **Q14**: Line 214: How did you determine the distractor is semantically irrelevant? Did you use an external VLM? Did you hand-curate such data points? Is there any reference (e.g., “point A”) that appears in both images?
> > > > >
> > > > > **A14**: Thank you for your correction. We agree that, under the mechanism of randomly concatenating images, we cannot guarantee that the sampled distractor image is completely semantically irrelevant to the original image; this is a more rigorous statement.
> > > > >
> > > > > We clarify our actual operation as follows: We indeed perform random concatenation of images, while preserving the original aspect ratio. We appreciate your detailed review and confirm that this description has been corrected in the revised manuscript.
> > > > >
> > > > > ---
> > > > >
> > > > > **Q15**: What is the rationale behind choosing the longest/shortest response for SFT/DPO?
> > > > >
> > > > > **A15**: Thank you for question. **We chose this approach primarily for the sake of simplicity and feasibility**, as we did not employ an external LLM to score the responses. We selected the longest generated response as the positive sample and the shortest generated response as the negative sample. This rationale is supported by the findings in instruction fine-tuning research. As noted by Zhao et al. [1], "Overall, our findings suggest that fine-tuning on the longest responses should be the default baseline for any work on instruction fine-tuning."
> > > > >
> > > > > [1] Zhao, Hao, et al. "Long is more for alignment: A simple but tough-to-beat baseline for instruction fine-tuning." ICML 2024.

---

> ### Comment · Reviewer_EuC5 · 2025-11-25
>
> Dear Authors,
>
> Thanks for the detailed response and thank you for incorporating much of the feedback I have provided. It seems like most of my concerns have been adequately addressed. However, I still do not see the detailed table in the appendix (which the authors mention to have been added in the updated manuscript). When including the table, could the authors also include the entries of the SOTA models (MM-eureka-7B, ThinkLite-7B, VL-Rethinker-7B)? Since the numbers are already present, this would serve as good baseline for comparison. I can come back once the manuscript has been updated again.

---

> > ### Author Response · Authors · 2025-11-25
> > **Response to Reviewer EuC5’s Follow-up Comment**
> >
> > Thanks for your follow-up comment. Since ICLR allows an extended manuscript of up to 10 pages, **we have moved the detailed table mentioned in A11 from the appendix into the main paper (Table 3 on page 7)**. Following your suggestion, **the detailed results for the comparison with advanced models are included in Appendix A.3 (Table 9 on page 16)** along with the corresponding explanations. In addition, the table referenced in A1 is now presented as Table 6 in the main paper (page 8), and the table mentioned in A4 appears in Appendix A.5 as Table 11 (page 17).
> >
> > **We appreciate your time and the detailed feedback again**, and if there are any further concerns, we are happy to discuss them.

---

> > > ### Author Response · Authors · 2025-11-28
> > >
> > > Dear Reviewer EuC5,
> > >
> > > I hope this message finds you well. Thanks for your follow-up comment again. As the discussion period is nearing its end with less than five days, I wanted to ensure we have addressed all your concerns satisfactorily. If there are any additional points or feedback you'd like us to consider, please let us know. Your insights are invaluable to us, and we're eager to address any remaining issues to improve our work.
> > >
> > > Thank you for your time and effort in reviewing our paper.

---

### Official Review · Reviewer_KowY · 2025-10-30

**Soundness:** 2
**Presentation:** 4
**Contribution:** 3
**Rating:** 6
**Confidence:** 5

**Summary:**

The authors introduces a fix that targets the fragility in MLLM-based visual reasoning. They introduce and incorporate Visual Perturbation (VP) during the post-training process e.g. GPRO. The results show that VP contributes to consistent improvements over multiple multi-modal visual reasoning benchmarks, such as MathVision and MathVerse.

**Strengths:**

I find the following aspects of this work remarkable

1. The authors have clearly demonstrated their motivation. The lack-of-robustness issue of existing reasoning MLLMs is well explained through clear examples such as Table 1.
2. The design of the experiments, along with all the verifications to demonstrate the effectiveness of VP, are comprehensive. I appreciate the authors’ effort to cover all the corners for as much as possible.

**Weaknesses:**

Still, I find several design flaws/loopholes with regard to VP. Out of the following two concerns, the first one is a major severe flaw that makes me question if the contribution of VP is genuine enough, especially if left unresolved.

1. **VP seems to introduce new problems, by rendering the original image unsolvable after perturbation.** Several more drastic perturbation strategies from VP, such as Random Crop 45%, may simply make original task unanswerable. For example, in Figure 3, after the applying the 45% Random Crop, the diagram in question no longer provides sufficient visual cues to solve x, for anyone (including human beings) that is concerned to start with. I believe the motivation of VP is to improve the robustness of MLLMs, but using unsolvable examples during post-training would achieve quite the opposite effect - the model would be driven to make ungrounded judgements, since now it’s fed with insufficient contexts to be based on.
2. **VP may seem to offer some improvements, but only up to a certain point.** From Table 2 & 3, it seems to me that the introduction of VP can only help improve the overall benchmarking performance up to a limited extent, all capped at 55%, suspiciously. To truly verify if VP can consistently improve benchmarking performances, as a suggestion, I would recommend the authors provide additional evidences, with incremental sizes in post training data, e.g. GRPO+VP with 2K, 4K, and 6K randomly sampled instances from GeoQA-8K.

**Questions:**

Please find my main concerns in the Weaknesses section. Other than that, I have one more minor concern.

1. It seems the authors have only tested with Qwen2.5-VL-7B as the only baseline/backbone. Have they tried deploying alternative baseline models, such as Qwen2.5VL-72B or variants of InternVL2/LLaVA? It would bring more contributive if we could see VP has generalizable deployability.

---

> ### Author Response · Authors · 2025-11-23
> **Response to Reviewer KowY**
>
> Thank you for your positive and encouraging feedback,  we would like to respond to your questions below.
>
> ---
>
> **Q1**: I believe the motivation of VP is to improve the robustness of MLLMs, but using unsolvable examples during post-training would achieve quite the opposite effect - the model would be driven to make ungrounded judgements, since now it’s fed with insufficient contexts to be based on.
>
> **A1**:Thank you for the thoughtful question. Our VP (Visual Perturbation) strategy is indeed designed to enhance the robustness of MLLMs but not by introducing unsolvable or information-deficient examples. Instead, each perturbed image in our approach preserves the original visual semantic content from the source diagrams (e.g., our mixup preserves the dominant visual features of the original image). The model still has access to the complete visual context required to answer the question.
> This simulates real-world scenarios where diagrams may appear in varied compositions (e.g., multi-panel figures, crowded slides, or split views), challenging the model’s ability to locate and reason over relevant visual elements.
>
> ---
>
> **Q2**: VP may seem to offer some improvements, but only up to a certain point. I would recommend the authors provide additional evidences, with incremental sizes in post training data, e.g. GRPO+VP with 2K, 4K, and 6K randomly sampled instances from GeoQA-8K.
>
> **A2**: We appreciate this insightful suggestion. We have already conducted the proposed scaling experiments. Specifically, we trained GRPO+VP models using 2K, 4K, and 6K randomly sampled and perturbed instances from GeoQA-8K. As shown in the table blow, from 2k to 4k, the improvement of VP increases from 1.19 to 1.67. From 4k to 8k, the improvement of VP increases from 1.67 to 1.96.
> |Datasets| Average Acc|
> |---|---|
> |GeoQA-2K| 51.95 |
> |GeoQA-2K + VP| 53.14 (+1.19)|
> |GeoQA-4K| 52.68 |
> |GeoQA-4K + VP|54.30 (+1.62)|
> |GeoQA-6K| 52.80|
> |GeoQA-6K + VP|54.47 (+1.67)|
> |GeoQA-8K| 52.77|
> |GeoQA-8K + VP| 54.73 (+1.96)|
>
> ---
>
> **Q3**: It seems the authors have only tested with Qwen2.5-VL-7B as the only baseline/backbone. Have they tried deploying alternative baseline models, such as Qwen2.5VL-72B or variants of InternVL2/LLaVA? It would bring more contributive if we could see VP has generalizable deployability.
>
> **A3**: Thank you for raising this important point regarding model generality. While our main results focus on Qwen2.5-VL-7B for controlled comparison and reproducibility, we are currently working on extending VP to Qwen3-VL (both 4B and 8B variants). Due to computation constraints, we are unable to report the final results in this round of the rebuttal. We commit to updating the latest results in a subsequent reply once the experiments are completed.

---

> > ### Comment · Reviewer_KowY · 2025-11-25
> > **The unsolvable are still hanging around, just in a different place.**
> >
> > I appreciate the clarifications on all my concerns. For the time being, I believe Q2 (Weakness #2) and Q3 (The Comment) have been adequately addressed.
> >
> > On Q1, upon further inspection, I believe my original understandings stem from two presentation issues that led to such confusion:
> >
> > 1. In Figure 2, **the 3 strategies involved in VP do not align up with the order that they are being introduced**. The correct order shall be, from top to bottom: I1 - dominance-perserving mix-up; I2 - random rotation; I3 - distractor concat.
> > 2. Figure 3 is presented in a way that **people could mistake those listed 10 strategies in Figure 3 as the actual perturbation strategies involved in the proposed VP method**. In fact, these 10 are only used in the ablation studies.
> >
> > *But still*, even though it's now clear that those in Figure 3 are not part of VP, the same unsolvability issue still lingers around, now in the ablation studies. In general, those drastic perturbation strategies (esp. *Random Crop 45%* and *Gaussian Noise 150*) can definitely make the original images unsolvable according to Figure 3. Therefore, their corresponding large performance drops in Table 7 (-11.5\% and -5.7\%) are certainly **less grounded baselines** for comparison purposes.
> >
> > (P.S. I notice there is a missing reference at Line 322 in the updated draft. You do not have to specifically address this typo in the follow-up comment. Let's simply focus on resolving my Q1.)

---

> > > ### Author Response · Authors · 2025-11-25
> > > **Response to Reviewer KowY’s Follow-up Comment on Q1**
> > >
> > > Thank you for your constructive feedback. **We sincerely appreciate the time and effort you've dedicated to this**, which have significantly improved the clarity of our manuscript. In response to your concerns regarding Q1, we have made the following revisions to enhance presentation and avoid potential confusion:
> > > 1. **We have aligned the ordering of all descriptions of our three Visual Perturbations (VP) throughout the paper to match Figure 2**, i.e., (1) dominance-preserving mixup, (2) random rotation and (3) distractor concatenation.
> > > This includes updates to the caption of Figure 2, the method descriptions in Section 4, the ordering of results in Tables 7 and 8, and all other textual references to these three strategies.
> > >
> > > 2. **We have revised Figure 3 to clearly separate our proposed VP methods from other ablation perturbations**. Specifically, all three VP strategies are now placed to the right of a dashed line, and the caption has been updated to boldly emphasize: “Visualization of different perturbation strategies used in our ablation studies,” making it clear that these perturbations in the left part are included solely for ablation analysis.
> > >
> > > 3. **We have removed the two disruptive ablation conditions**—Random Crop 45% and Gaussian Noise (std=150). These were initially included to highlight our semantic-preserving rules, but we have realized that both perturbations make the images nearly unsolvable, which is inappropriate for fair comparison. **In addition, we have added a new Color Shift ablation and reorganized the content of ablation study accordingly**.
> > >
> > > Once again, thank you for your detailed comments. We firmly believe it is our responsibility to eliminate any ambiguity that might confuse readers, and your suggestions have substantially improved the quality of our manuscript. We are truly grateful for your time and thoughtful engagement with our work. If any concerns remain, we warmly welcome further discussion.

---

> > > > ### Comment · Reviewer_KowY · 2025-11-25
> > > > **Much better now.**
> > > >
> > > > I can finally call it closure as all my original concerns have been properly addressed in the new updated draft. That being said, I am happy to raise my rating to 8.

---

> > > > > ### Author Response · Authors · 2025-11-25
> > > > > **Thanks for raising your score**
> > > > >
> > > > > We are very glad to hear that all of your concerns have been addressed. Thank you for your thoughtful review and constructive suggestions, which have greatly improved the quality and clarity of our manuscript. We sincerely appreciate the time and effort you've dedicated to this. Thanks again for your review and comments.

---

### Official Review · Reviewer_ixh4 · 2025-10-31

**Soundness:** 3
**Presentation:** 3
**Contribution:** 2
**Rating:** 6
**Confidence:** 4

**Summary:**

The authors study how well multimodal LLMs actually use visual information in math-reasoning tasks. Their experiments build on 2 primary findings: (a) On vision tasks, a language-only model given generated image captions can improve over the multimodal model that produced the captions. (b) Multimodal models are susceptible to image rotation. To mitigate this issue, the authors propose a data augmentation based training, that improves model performance across all the tasks. The

**Strengths:**

The main strength of the paper lies in its motivating experiment that shows the key weaknesses they are targeting to. By showing strong evidence that current MLLMs don't reliably use visual information, they motivate the necessity of augmentation based training. The proposed algorithm uses conventional augmentation methods that visual perturbation and improve the model's performance across $4$ benchmarks. Furthermore, the authors conduct extensive study on which perturbation method affects each benchmark the most. Overall, the authors provide a comprehensive empirical study showing the strengths of their proposed training algorithm.

**Weaknesses:**

There are few questions regarding the experiment setting that I would like the authors to address.



a) How does the fine-tuning of the vision tower affect the performance? Is the performance gain primarily because of the weakness in the vision tower? What happens if you freeze the vision tower or the language model during training?

b) Can perturbations be applied at evaluation time to improve the model performance further? That is, one could apply different kinds of perturbations to the image and then take a majority vote among all responses of the model to the different perturbed images.

c) How do the different perturbation types affect the model performance when the same (or other) perturbations are applied to the images during evaluation? And can the authors provide more quantitative arguments on why such perturbations helped the model after training?

**Questions:**

Please check my questions above.

---

> ### Author Response · Authors · 2025-11-23
> **Response to Reviewer ixh4**
>
> Thank you for review and insightful questions. We would like to respond your questions point by point below.
>
> ---
>
> **Q1**: How does the fine-tuning of the vision tower affect the performance? Is the performance gain primarily because of the weakness in the vision tower? What happens if you freeze the vision tower or the language model during training?
>
> **A1**: Thank you for your insightful question regarding the role of Vision Tower (VT) fine-tuning. Our results demonstrate that the performance gain is not primarily due to addressing a weak VT. Specifically, on Geometry-3k, the difference between freezing and unfreezing the VT is negligible: the GRPO baseline achieved 53.0% (frozen) vs. 53.1% (unfrozen), and GRPO + VP achieved 54.7% (frozen) vs. 54.9% (unfrozen). This minimal variation (0.1–0.2%) proves the VT is already robust. Consequently, we confirm that VP acts as for regularization focused downstream on the MLP-based Vision-Language Merger(connector), enhancing the stability and alignment of features consumed by the Language Model. This phenomenon aligns with the established paradigm for large-scale VLMs; as noted in the Qwen2.5-VL report[1], their post-training, including SFT and DPO, is performed with the ViT parameters frozen. We will include this additional experiment in the Appendix.
>
> |Freeze Vision Tower|Method|avg|
> |---|---|---|
> |Unfreeze|GRPO|53.1|
> |Freeze|GRPO|53.0|
> |Unfreeze|GRPO + VP|54.9|
> |Freeze|GRPO + VP |54.7|
>
> [1] Bai, Shuai, et al. "Qwen2. 5-vl technical report." arXiv preprint arXiv:2502.13923 (2025).
>
> ---
>
> **Q2**: Can perturbations be applied at evaluation time to improve the model performance further? That is, one could apply different kinds of perturbations to the image and then take a majority vote among all responses of the model to the different perturbed images.
>
> **A2**: Thank you for raising this point, which falls under the scope of Test-Time Augmentation or scaling. This is indeed a very promising research direction worth further exploration. However, if we were to use $K$ different perturbations for majority voting, the results would need to be compared against the results of $K$ times answers with clean image. Due to time constraints, it is hard for us to surpass this baseline yet, but we will continue to research this in depth in the future.
>
> ---
>
> **Q3**: How do the different perturbation types affect the model performance when the same (or other) perturbations are applied to the images during evaluation? And can the authors provide more quantitative arguments on why such perturbations helped the model after training?
>
> **A3**: Thank you for this important question. To more thoroughly understand how different perturbation types influence model performance under matched or mismatched perturbations at evaluation time, we are currently conducting an additional analysis based on visual feature distance measurement. Concretely, we extract visual embeddings(after alignment) for clean images and for each perturbation type, and compute quantitative metrics such as cosine distance, L2 distance, and MMD between the corresponding feature distributions. This analysis aims to characterize how each perturbation affects the internal visual representation space and how the representation shift relates to robustness during evaluation. These experiments are in progress, and we will include the full quantitative results in the next round.

---

### Official Review · Reviewer_oSMF · 2025-11-03

**Soundness:** 2
**Presentation:** 2
**Contribution:** 2
**Rating:** 2
**Confidence:** 3

**Summary:**

This paper studies how MLLM models utilize its visual inputs and how robust these models are to input perturbations. It proposes to use different input perturbation methods to input images when training MLLMs. Experiment results show modest improvements in performances.

**Strengths:**

1. The paper is generally clearly written but with some logical jump (please see Q2 in weakness).

2. I appreciate the authors' motivation in conducting analysis experiments in Section 3. Such experiments can help us understand where current MLLMs fall short, e.g., whether they cannot perceive the visual inputs well enough or it's their lack of reasoning capability, or even that their reasoning is not well grounded on the inputs (more in weakness Q1).

**Weaknesses:**

1. The motivation in the experiments conducted in Figure 1 is interesting, but I feel the conclusion that "MLLMs may generate
accurate visual descriptions but fail to effectively integrate them during reasoning" is not very well supported by the experiment setup. Specifically, Answer C has better performance than Answer B does not necessary imply MLLMs fail to integrate visual description during reasoning. It might be from the fact that the explicitly produced the caption may help ground models to the input image in answering the question. To support the claim, shouldn't the setup be replacing QwenLM with Qwen-VL in the Answer C setup? This may help us understand where MLLMs fall short.

2. In the paper, there is logical jump from MLLM's reasoning capability based on visual inputs to their robustness to visual input perturbations. It's not clear to me where the connection is, and how improving robustness to input perturbation helps reasoning. These seems to me as orthogonal axes.

3. The main method proposed in the paper appears marginal, which applies existing input perturbation (data augmentation) methods to input images in model training pipeline. It is thus not surprising to see that augmentation helps improves model performance, but also only with relatively small gains (as seen in Table 2).

**Questions:**

Could the authors clarify the main contribution of the paper and how that relates to existing data augmentation literature?

---

> ### Author Response · Authors · 2025-11-23
> **Response to Reviewer oSMF (1/4)**
>
> Thanks a lot for your review and detailed feedback! We would like to address your concerns below.
>
> ---
>
> **Q1**:"The motivation in the experiments conducted in Figure 1 is interesting, but I feel the conclusion that "MLLMs may generate accurate visual descriptions but fail to effectively integrate them during reasoning" is not very well supported by the experiment setup. Specifically, Answer C has better performance than Answer B does not necessary imply MLLMs fail to integrate visual description during reasoning. It might be from the fact that the explicitly produced the caption may help ground models to the input image in answering the question. To support the claim, shouldn't the setup be replacing QwenLM with Qwen-VL in the Answer C setup? This may help us understand where MLLMs fall short."
>
> **A1**: Thank you for this insightful suggestion. We would like to address this by first explaining the rationale behind our original experimental design and then presenting the new results following your advice.
>
> First, regarding our motivation, we hypothesize that a caption-augmented language model establishes a natural lower bound for the performance of an ideal multimodal model on visual reasoning tasks, under the assumption that both models possess comparable language understanding capabilities. Since image captions are compressed representations of visual content, they inherently contain less information than the original images. **Thus, a well-aligned and effective MLLM, which can directly access and process raw visual inputs, should in principle outperform or at least match a language model that only relies on generated captions.** When this expectation is not met, it suggests that the MLLM may be underutilizing visual information.
>
> Second, we believe our original comparison between Qwen-VL (Answer B) and QwenLM (Answer C) was reasonable under this hypothesis. **As stated in Table 4 (Linguistic Performance) in the Qwen2.5-VL Technical Report [1], the LLM of Qwen-VL is initialized from the QwenLM backbone and its capabilities of the LLM model are maintained to be equivalent to the base QwenLM.**
>
> Finally, we fully agree with you that replacing QwenLM with Qwen-VL in the Answer C setup provides a more rigorous control to strictly isolate the "integration" factor. As suggested, **we conducted the additional experiment where we feed the exact same "Question + Caption" input to Qwen-VL to get a new Answer C**. As shown in the table below, even when using the exact same model, the setup with generated captions (Answer C - New) still outperforms the image-only setup (Answer B).
>
> |Answer A|Answer B|Answer C|Answer C (New)|
> | --- | --- | --- | --- |
> |24.3|25.6|28.8|29.0|
>
> [1] Bai, Shuai, et al. "Qwen2. 5-vl technical report." arXiv preprint arXiv:2502.13923 (2025).

---

> ### Author Response · Authors · 2025-11-23
> **Response to Reviewer oSMF (2/4)**
>
> **Q2**:"In the paper, there is logical jump from MLLM's reasoning capability based on visual inputs to their robustness to visual input perturbations. It's not clear to me where the connection is, and how improving robustness to input perturbation helps reasoning. These seems to me as orthogonal axes."
>
> **A2**: Thank you for raising this question. We clarify that there is a connection between robustness and reasoning capability, and below is our detailed explanation:
> First, we need to reiterate our research motivation and methodological approach. Our study began with an  analysis of existing MLLM limitations (as discussed in your Q1). This analysis revealed that MLLMs currently fail to effectively integrate and rationally apply visual input information in mathematical reasoning tasks. Given this challenge, we conducted a literature review and found that traditional visual data augmentation methods are difficult to directly apply to MLLMs effectively. Therefore, based on the semantic preserving principle, we  designed three simple visual perturbation methods and validated their effectiveness in MLLMs training.
>
> Second, **we want to highlight that our focus is on improving the visual abilities of MLLMs, and we specifically choose visual reasoning because it is highly challenging as noted in various surveys** [1,2,3]. And we clarify that the mathematical reasoning task is only one task used to validate the effectiveness of Visual Perturbation (VP), and the logical connection is not from visual perturbation to reasoning performance. **To further demonstrate the effectiveness of VP, we provide follow-up experiments on non-math reasoning tasks here**. We validate that our method is equally effective on natural images and conducted experiments using the SegZero training framework [4] (SegZero and VisionReasoner [5] are high-impact works on GitHub that form a series—SegZero provides the training codebase while VisionReasoner offers a more advanced model). We applied VP training to the VisionReasoner model on natural image segmentation tasks:
> | Method | ReasonSeg val | ReasonSeg test | RCO test | ARCO+ test | ARCOg test |
> | --- | --- | --- | --- | --- | --- |
> | VisionReasoner-7B | 66.3 | 63.6 | 78.9 | 74.9 | 71.3 |
> | VisionReasoner-7B + VP | 67.1 | 65.2 | 80.4 | 74.1 | 72.9 |
>
> Due to time constraints, we were unable to fully optimize the performance on this task, yet the results still robustly demonstrate the effectiveness of VP.
> Our ultimate goal is for VP to become a consensus in the MLLM community, offering the possibility of further performance improvement without requiring additional data, or complex algorithmic and model structure design.
>
> [1] Wang, Yaoting, et al. "Multimodal chain-of-thought reasoning: A comprehensive survey." arXiv preprint arXiv:2503.12605 (2025).
>
> [2] Su, Zhaochen, et al. "Thinking with images for multimodal reasoning: Foundations, methods, and future frontiers." arXiv preprint arXiv:2506.23918 (2025).
>
> [3] Bi, Jing, et al. "Why reasoning matters? a survey of advancements in multimodal reasoning (v1)." arXiv preprint arXiv:2504.03151 (2025).
>
> [4] Liu, Yuqi, et al. "Seg-Zero: Reasoning-Chain Guided Segmentation via Cognitive Reinforcement"
>
> [5] Liu, Yuqi, et al. "VisionReasoner: Unified Visual Perception and Reasoning via Reinforcement Learning"

---

> ### Author Response · Authors · 2025-11-23
> **Response to Reviewer oSMF (3/4)**
>
> **Q3**:"The main method proposed in the paper appears marginal, which applies existing input perturbation (data augmentation) methods to input images in model training pipeline. It is thus not surprising to see that augmentation helps improves model performance, but also only with relatively small gains (as seen in Table 2)."
>
> **A3**: Thank you for this critical assessment. We respectfully address these points by clarifying our method’s contribution and contextualizing our performance gains through relevant improvement.
>
> **We first clarify that we are not merely applying existing perturbation methods.** While the components are simple, our contribution is the analysis-driven design rooted in the semantic preserving principle, aimed specifically at resolving the MLLM's visual integration failure (discussed in Q1). On the contrary, simply using naive augmentation (the "existing methods" implied) often causes performance drops in MLLMs (as noted in our ablation studies).
>
> **Furthermore, we clarify that the perceived "relatively small gains" are misunderstanding.** Our simple, data-efficient method achieves 1-2 percentage points improvement over the GRPO baseline (in Table 2), which is competitive with algorithmic methods presented recently [1, 2, 3]. Crucially, if we contextualize our performance against the typical SOTA reporting standard (comparing Base Model to Our Method + GRPO Training) [4, 5, 6], our average improvement is substantial, reaching 5 percentage points across benchmarks. Our method's value lies in its simplicity, broad applicability, and demonstrated competitive performance.
>
> | Perturbation Type | Average|
> |---|---|
> | None (Baseline) | 53.1 |
> | Gaussian Blur | 49.0 (–7.8%)|
> | Random Crop 15% | 50.4 (–5.1%) |
> | Gaussian noise (std=50) | 51.9 (–2.3%) |
> | Standard Mixup | 51.2 (–3.6%) |
> | Color Shift  (HSV ±0.1) | 53.2 (+0.2%) |
>
> [1] Hu, Yijie, et al. "Can MLLMs Absorb Math Reasoning Abilities from LLMs as Free Lunch?." NeurIPS 2025
>
> [2] Luo, Ruilin, et al. "Unlocking Multimodal Mathematical Reasoning via Process Reward Model." NeurIPS 2025
>
> [3] Cao, Qi, et al. "DreamPRM: Domain-Reweighted Process Reward Model for Multimodal Reasoning." NeurIPS 2025
>
> [4] Yao, Huanjin, et al. "R1-ShareVL: Incentivizing Reasoning Capability of Multimodal Large Language Models via Share-GRPO." NeurIPS 2025
>
> [5] Wang, Xiyao, et al. "Sota with less: Mcts-guided sample selection for data-efficient visual reasoning self-improvement." NeurIPS 2025
>
> [6] Xiao, Wenyi, and Leilei Gan. "Fast-Slow Thinking GRPO for Large Vision-Language Model Reasoning." NeurIPS 2025

---

> ### Author Response · Authors · 2025-11-24
> **Response to Reviewer oSMF (4/4)**
>
> **Q4**: Could the authors clarify the main contribution of the paper and how that relates to existing data augmentation literature?
>
> **A4**: Thank you for your question. We appreciate the opportunity to clarify the main contribution of our work and its relationship to existing data augmentation literature.
> **Our paper can be summarized as follows**:
> - Our work is motivated by a key insight: “Better reasoning begins with better seeing.”
> - We make two key observations: (1) caption-augmented LLMs can match or even surpass MLLMs, and (2) minor image perturbations lead to significant accuracy declines—revealing fundamental weaknesses in how current MLLMs integrate visual information during reasoning.
> - We propose Visual Perturbation (VP), a simple and lightweight method that is broadly applicable across training paradigms.
> - Comprehensive experiments demonstrate VP’s consistent effectiveness across diverse datasets, models, and training setups.
> - Ablation studies further reveal that perturbation effects vary by task.
>
> **Our work fundamentally differs from existing data augmentation approaches in one key aspect: existing approaches do not specifically target the emerging class of multimodal reasoning tasks for MLLMs**. Below we clarify the relationship and distinctions in detail:
>
> (1) MixGen [1] and RobustMixGen [2] generate new **image–text pairs** through interpolation or mixing to improve generalization or robustness under distribution shift. In contrast, VP does not synthesize new samples; it applies structured perturbations to existing images while preserving their original labels and semantics.
>
> (2) XTRA [3] and LEMDA [4] operate at the feature or retrieval level, enhancing training **via cross-modal retrieval or learned feature-space augmentations**. VP, by contrast, is a purely input-level, image-space method that requires no external data, retrieval system, or representation learning.
>
> (3) Verma et al. [5] evaluate MLLMs under various image and language perturbations, but they **focus on simpler VQA** tasks (e.g., answering “Yes” or “No”). VP, by contrast, targets the more challenging tasks of visual reasoning.
>
> (4) NoiseBoost [6] alleviates hallucinations in MLLMs through the integration of noise feature perturbations, but its evaluation is **limited to hallucination mitigation**. In contrast, VP focuses on visual reasoning and is designed with semantics-preserving rules: we maintain the core visual content while adding perturbations to images.
>
> [1] Hao, Xiaoshuai, et al. "Mixgen: A new multi-modal data augmentation." WACV 2023.
>
> [2] Kim, Sunwoo, et al. "RobustMixGen: Data augmentation for enhancing robustness of visual–language models in the presence of distribution shift." Neurocomputing 619 (2025): 129167.
>
> [3] Gur, Shir, et al. "Cross-modal retrieval augmentation for multi-modal classification." arXiv preprint arXiv:2104.08108 (2021).
>
> [4] Liu, Zichang, et al. "Learning multimodal data augmentation in feature space." ICLR 2023.
>
> [5] Verma, Aayush Atul, et al. "Evaluating multimodal large language models across distribution shifts and augmentations." CVPRW 2024.
>
> [6] Wu, Kai, et al. "Noiseboost: Alleviating hallucination with noise perturbation for multimodal large language models." arXiv preprint arXiv:2405.20081 (2024).

---

> > ### Author Response · Authors · 2025-11-28
> >
> > Dear Reviewer oSMF,
> >
> > I hope this message finds you well. As the discussion period is nearing its end with less than five days, I wanted to ensure we have addressed all your concerns satisfactorily. If there are any additional points or feedback you'd like us to consider, please let us know. Your insights are invaluable to us, and we're eager to address any remaining issues to improve our work.
> >
> > Thank you for your time and effort in reviewing our paper.

---

### Author Response · Authors · 2025-12-01
**Rebuttal Summary for Area Chair**

Dear AC and all reviewers,

Thank you for your valuable time and insightful feedback. We sincerely appreciate the constructive engagement throughout the discussion period, which has significantly strengthened our work.

---

Our initial scores were 2, 6, 6, 4. Before the reviewer response period was frozen, two reviewers provided feedback.
- Reviewer KowY (score 6 -> score 8, confidence 5) mentioned that we had **addressed all of their concerns** and would **raise their score to 8** during the early stage of rebuttal.
- Reviewer EuC5 (score 4, confidence 4) mentioned that we **had almost addressed all of their concerns**; the **only remaining clarification point** concerned a detailed table we mentioned adding in the Appendix. We explained that this was because we had placed this table in the main paper. Due to the discussion closed, the reviewer was unable to provide a further response. We are confident that our rebuttal resolves all concerns and anticipate a potential upgrade of their score to 6.

The remaining reviewers, Reviewer ixh4 (score 6 confidence 4) with a positive rating and Reviewer oSMF (score 2 confidence 3) with a negative rating did not provide a further response. Here, we would like to highlight our explanation for all the questions raised by Reviewer oSMF (score 2, confidence 3), whose concerns, given with low confidence, stemmed from a misunderstanding of our experiments and motivation. These questions are actually easy to answer:

**1.** The motivation in the experiments conducted in Figure 1 is interesting, but to better support the claim, **the setup should be replacing QwenLM with Qwen-VL in the Answer C setup**.

- **Our original comparison** between Qwen-VL (Answer B) and QwenLM (Answer C) **was reasonable** under this hypothesis. As stated in Table 4 (Linguistic Performance) in the Qwen2.5-VL Technical Report [1], the LLM of Qwen-VL is initialized from the QwenLM backbone and its capabilities of the LLM model are maintained to be equivalent to the base QwenLM.
- **We conducted the suggested experiment**. Even when using the exact same model, the setup with generated captions **still outperforms** the image-only setup.

**2.** There is **logical jump** from MLLM's reasoning capability to their robustness to visual input perturbations.

- **This is not a logical jump**. We want to highlight that our focus is on improving the visual abilities of MLLMs, and we clarify that the mathematical reasoning task is one of tasks used to validate the effectiveness of Visual Perturbation (VP). We specifically choose visual reasoning because **it is highly challenging as noted in various surveys**.
- To further demonstrate the effectiveness of VP, **we provide follow-up experiments** on the other task.

**3.** The main method proposed in the paper appears marginal, which applies **existing** input perturbation methods to input images  and achieves **relatively small gains**.

- **We are not merely applying existing perturbation methods**. While the components are simple, the analysis-driven design rooted in the semantic preserving principle. On the contrary, **simply using** naive perturbation (the **"existing methods"** implied) often caused **performance drops** as noted in our ablation studies.
- **"Relatively small gains" is a misunderstanding**. Our method achieves 1-2 percentage points improvement over the GRPO baseline (in Table 2), which is **competitive with algorithmic methods presented recently** [2, 3, 4] (accepted by **NeurIPS 2025**). Crucially, if we contextualize our performance against the typical SOTA reporting standard (comparing Base Model to Our Method + GRPO Training) [5, 6, 7] (accepted by **NeurIPS 2025**), our average improvement is substantial, reaching 5 percentage points across benchmarks.

Since the single negative score was due to a fundamental misunderstanding of our work, we are confident that Reviewer oSMF will **reconsider our work and at least raise the score to 4**. Overall, we strongly believe that **had the scores not been locked, our final ratings would have reached 4, 6, 8, 6 (raising the average to 6)**.


[1] Bai, Shuai, et al. "Qwen2. 5-vl technical report." arXiv preprint arXiv:2502.13923 (2025).

[2] Hu, Yijie, et al. "Can MLLMs Absorb Math Reasoning Abilities from LLMs as Free Lunch?." NeurIPS 2025

[3] Luo, Ruilin, et al. "Unlocking Multimodal Mathematical Reasoning via Process Reward Model." NeurIPS 2025

[4] Cao, Qi, et al. "DreamPRM: Domain-Reweighted Process Reward Model for Multimodal Reasoning." NeurIPS 2025

[5] Yao, Huanjin, et al. "R1-ShareVL: Incentivizing Reasoning Capability of Multimodal Large Language Models via Share-GRPO." NeurIPS 2025

[6] Wang, Xiyao, et al. "Sota with less: Mcts-guided sample selection for data-efficient visual reasoning self-improvement." NeurIPS 2025

[7] Xiao, Wenyi, and Leilei Gan. "Fast-Slow Thinking GRPO for Large Vision-Language Model Reasoning." NeurIPS 2025

---

### Meta-Review · Area_Chair_ZRcr · 2026-01-11

**Summary:**

The main concerns from the reviewers are following:

- Reviewer **oSMF**:
  - **W1**: The motivation in the experiments conducted in Figure 1 is interesting, but I feel the conclusion that "MLLMs may generate accurate visual descriptions but fail to effectively integrate them during reasoning" is not very well supported by the experiment setup.
  - **W2**: There is logical jump from MLLM's reasoning capability based on visual inputs to their robustness to visual input perturbations.
  - **W3**: The main method proposed in the paper appears marginal. The performance gain in the experiments is limited.

- Reviewer **ixh4**:
  - **W1**: Several questions exist regarding the experiment settings.

- Reviewer **KowY**:
  - **W1**: The proposed approach seems to introduce new problems, by rendering the original image unsolvable after perturbation.

- Reviewer **EuC5**:
  - **W1**: The reviewer proposes a number of regarding the soundness of experimental setup and result analysis.

**Reviewer Concerns:**

- **Concerns addressed in the rebuttal:**
  - Overall, I believe that the authors have conducted a good job to address the technical concerns of Reviewer **ixh4**, **KowY**, and **zkVT**.

- **Concerns remained outstanding:**
  - I think **W2** and **W3** of Reviewer **oSMF** are not fully addressed. Considering the key motivation of the paper: current MLLMs may perceive visual content but fail to effectively integrate it during reasoning, I agree with this point, but this finding is not original, but a common motivation for recent researches on solving visual reasoning with visual prompting techniques and the paradigm of "thinking with images". This shows that visual perturbation is not a unique solution for this problem. While the paper lacks discussions as well as experimental comparisons to strategies beyond visual perturbation. Furthermore, in my understanding, the major technical contribution of the paper lies in discovering a set of effective visual perturbation techniques out of all. For this contribution being significant, it is necessary to verify whether they can fit with broad and general visual reasoning tasks. While the experiments mainly focus on math reasoning problems, which are a relatively narrow subset.

**Reviewer Scores:**

As discussed in the Reviewer Concerns, the technical concerns are indeed addressed by the author responses. I believe that Reviewer **ixh4**, **KowY**, and **zkVT** would hold positive evaluations on the paper. While in my view, the remaining concerns (**W2** and **W3** of Reviewer **oSMF**) are major issues that not ignorable for evaluating the quality of the current submission.

---

### Decision · Program_Chairs · 2026-01-26

Reject